# FedGT: Federated Node Classification with Scalable Graph Transformer

## Abstract

Graphs are widely used to model relational data. As graphs are getting larger and larger in real-world scenarios, there is a trend to store and compute subgraphs in multiple local systems. For example, recently proposed *subgraph federated learning* methods train Graph Neural Networks (GNNs) distributively on local subgraphs and aggregate GNN parameters with a central server. However, existing methods have the following limitations: (1) The links between local subgraphs are missing in subgraph federated learning. This could severely damage the performance of GNNs that follow message-passing paradigms to update node/edge features. (2) Most existing methods overlook the subgraph heterogeneity issue, brought by subgraphs being from different parts of the whole graph. To address the aforementioned challenges, we propose a scalable **Fed**erated **G**raph **T**ransformer (**FedGT**) in the paper. Firstly, we design a hybrid attention scheme to reduce the complexity of the Graph Transformer to linear while ensuring a global receptive field with theoretical bounds. Specifically, each node attends to the sampled local neighbors and a set of curated global nodes to learn both local and global information and be robust to missing links. The global nodes are dynamically updated during training with an online clustering algorithm to capture the data distribution of the corresponding local subgraph. Secondly, FedGT computes clients' similarity based on the aligned global nodes with optimal transport. The similarity is then used to perform weighted averaging for personalized aggregation, which well addresses the data heterogeneity problem. Finally, extensive experimental results on 6 datasets and 2 subgraph settings demonstrate the superiority of FedGT.

## 1 Introduction

Many real-world relational data can be represented as graphs, such as social networks (Fan et al., 2019), molecule graphs (Satorras et al., 2021), and commercial trading networks (Xu et al., 2021). Due to the ever-growing size of graph (Hu et al., 2020a) and stricter privacy constraints such as GDPR (Voigt & Von dem Bussche, 2017), it becomes more practical to collect and store sensitive graph data in local systems instead in a central server. For example, banks may have their own relational databases to track commercial relationships between companies and customers. In such scenarios, it is desirable to collaboratively train a powerful and generalizable graph mining model for business, e.g., loan prediction with distributed subgraphs while not sharing private data. To this end, subgraph federated learning (Zhang et al., 2021; Wu et al., 2021; 2022a; Xie et al., 2022; He et al., 2021) has been recently explored to resolve the information-silo problem and has shown its advantage in enhancing the performance and generalizability of local graph mining models such as Graph Neural Networks (GNNs) (Wu et al., 2020; Zhou et al., 2020).

However, subgraph federated learning has brought unique challenges. Firstly, different from federated learning in other domains such as CV and NLP, where data samples of images and texts are isolated and independent, nodes in graphs are connected and correlated. Therefore, there are potentially missing links/edges between subgraphs that are not captured by any client (illustrated in Figure 1), leading to severe information loss or bias. This becomes even worse when GNNs are used as graph mining models because most GNNs follow a message-passing scheme that aggregates node embeddings along edges (Gilmer et al., 2017; Hamilton et al., 2017a). To tackle this problem, existing works try to recover missing neighborhood information with neighborhood generator (Zhang et al., 2021; Peng et al., 2022) or share node embeddings of local subgraphs among clients (Chen et al.,

2021). However, these solutions either struggle to fully recover the missing links or bring risk to data privacy protection.

Another important challenge in subgraph federated learning is data heterogeneity which is also illustrated in Figure 1. Some subgraphs may have overlapping nodes or similar label distributions (e.g., client subgraph 2&3 in Figure 1). On the contrary, some subgraphs may be completely disjoint and have quite different properties (e.g., client subgraph 1&3 in Figure 1). Such a phenomenon is quite common in the real world since the data of clients may come from different domains and comprise different parts of the whole graph. However, most of the existing methods (Zhang et al., 2021; Wu et al., 2021; He et al., 2021) fail to consider the data heterogeneity issue and may lead to sub-optimal performance for each client. More examples and discussions of the challenges in subgraph federated learning are included in the Appendix. B.

To address the aforementioned challenges, we propose a scalable **F**ederated **G**raph **T**ransformer (**FedGT**) in this paper. Firstly, in contrast to GNNs that follow message-passing schemes and focus on local neighborhoods, Graph Transformer has a global receptive field to learn long-range dependencies and is therefore more robust to missing links. However, the quadratic computational cost of the vanilla transformer architecture inhibits its direct application in the subgraph FL setting. In FedGT, a novel hybrid attention scheme is proposed to bring the computational cost to linear with theoretically bounded approximation error. Specifically, in the computation of clients, each node attends to the sampled neighbors in the local subgraph and a set of curated global nodes representing the global context. The global nodes are dynamically updated during the training of FedGT with an online clustering algorithm and serve the role of supplementing missing information in the hybrid attention. Secondly, to tackle the data heterogeneity issue, FedGT leverages a personalized aggregation scheme that performs weighted averaging based on the estimated similarity matrix between clients. The similarity is calculated based on global nodes from different clients since they can reflect their corresponding data distribution of local subgraphs. Since there are no fixed orders in the set of global nodes, we apply optimal transport to align two sets of global nodes before calculating clients' similarity. To further protect the privacy of local clients, we also apply local differential privacy (LDP) techniques. Finally, extensive experiments on 6 datasets and 2 subgraph settings demonstrate that FedGT can achieve state-of-the-art performance.

## 2 RELATED WORK

### 2.1 FEDERATED LEARNING AND FEDERATED GRAPH LEARNING

Federated Learning (FL) (McMahan et al., 2017a; Yang et al., 2019; Arivazhagan et al., 2019; T Dinh et al., 2020; Fallah et al., 2020; Beaussart et al., 2021) is an emerging collaborative learning paradigm over decentralized data. Specifically, multiple clients (e.g., edge data centers, banks, companies) jointly learn a machine learning model (i.e., global model) without sharing their local private data with the cloud server or other clients. However, different from the commonly studied image and text data in FL, graph-structure data is correlated and brings unique challenges to FL such as missing cross-client links. Therefore, exploring federated graph learning approaches that tackle the unique challenges of graph data is required.

Federated Graph Learning (FGL) (Wang et al., 2022) can be classified into graph and subgraph FL according to the types of graph tasks. Graph FL methods (Xie et al., 2021b; He et al., 2022) assume that different clients have completely disjoint graphs (e.g., molecular graphs), which is similar to common federated learning tasks on CV (e.g., federated image classification). For example, GCFL+ (Xie et al., 2021b) proposes a graph clustered federated learning framework to deal with the data heterogeneity in graph FL. It dynamically bi-partitions a set of clients based on the gradients of GNNs. On the contrary, in subgraph FL (Zhang et al., 2021; Wu et al., 2021; 2022a; Peng et al., 2022; He et al., 2021; Baek et al., 2023), each local client holds a subgraph that belongs to the whole large graph (e.g., social network). The subgraphs are correlated and there may be missing links across clients. To tackle the unique challenges in subgraph FL, FedSage+ and FedNI (Zhang et al., 2021; Peng et al., 2022) utilize neighbor generators to recover missing cross-client links and neighbor nodes. However, they only focus on immediate neighbors and can hardly recover all the missing information. On the other hand, FedGraph (He et al., 2021) augments local data by requesting node information from other clients, which may compromise data privacy constraints. In this paper, we focus on the subgraph FL for node classification. Unlike existing GNN-based methods, our method

tackles the problem from a completely different perspective by leveraging powerful and scalable Graph Transformer architectures.

## 2.2 GRAPH NEURAL NETWORK AND GRAPH TRANSFORMER

Graph Neural Networks (GNNs) (Kipf & Welling, 2017; Hamilton et al., 2017a; Han et al., 2021; Zhang et al., 2019; Bojchevski et al., 2020; Jin et al., 2020) are widely used in a series of graph-related tasks such as node classification and link prediction. Typically, GNNs follow a message-passing scheme that iteratively updates the node representations by aggregating representations from neighboring nodes. However, GNNs are known to have shortcomings such as over-smoothing (Chen et al., 2020) and over-squashing (Topping et al., 2021), which are further exacerbated in the subgraph FL setting where cross-client links are missing.

In recent years, Graph Transformer (Ying et al., 2021; Kreuzer et al., 2021) has shown its superiority in graph representation learning. Most works of Graph Transformer focus on graph classification on small graphs (Ying et al., 2021; Kreuzer et al., 2021), where each node is regarded as a token and special positional encodings are designed to encode structural information. For instance, Graphormer (Ying et al., 2021) uses centrality, edge, and hop-based encodings and achieves state-of-the-art performance on molecular property prediction tasks. However, the all-pair nodes attention scheme has a quadratic complexity and is hard to scale to large graphs. Recently, some works attempt to alleviate the quadratic computational cost and build scalable Graph Transformers based on node sampling or linear approximation (Dwivedi & Bresson, 2020; Zhang et al., 2020; Zhao et al., 2021; Wu et al., 2022b; Zhang et al., 2022; Rampášek et al., 2022; Chen et al., 2023). However, most of these methods sacrifice the advantage of global attention in the Transformer architecture and can hardly extend to the more challenging subgraph FL setting due to issues such as data heterogeneity.

## 3 PRELIMINARIES

### 3.1 PROBLEM DEFINITION

**Node Classification.** We first introduce node classification in the centralized setting. Consider an unweighted graph $G = (A, X)$ where $A \in \mathbb{R}^{n \times n}$ represents the symmetric adjacency matrix with $n$ nodes, and $X \in \mathbb{R}^{n \times p}$ is the attribute matrix of $p$ attributes per node. The element $A_{ij}$ in the adjacency matrix equals 1 if there exists an edge between node $v_i$ and node $v_j$, otherwise $A_{ij} = 0$. The label of node $v_i$ is $y_i$. In the semi-supervised node classification problem, the classifier (e.g., GNN, graph transformer) has the knowledge of $G = (A, X)$ and a subset of node labels. The goal is to predict the labels of the other unlabeled nodes by learning a classifier.

**Subgraph Federated Learning.** In subgraph FL, there is a central server $S$ coordinating the training process, and a set of local clients $\{C_1, C_2 \ldots C_M\}$ taking part in the training. $M$ is the number of clients. Each local client stores a subgraph $G_i = (A_i, X_i)$ of the global graph $G$ with $n_i$ nodes. Two subgraphs from different clients might have overlaps or are completely disjoint. Note that there might be edges between two subgraphs in the whole graph but are not stored in any client (missing links). The central server $S$ only maintains a graph mining model but stores no graph data. Each client $C_i$ cannot directly query or retrieve data from other clients due to privacy constraints.

**Training Goal.** The goal of subgraph federated learning is to leverage isolated subgraph data stored in distributed clients and collaboratively learn node classifiers $\mathcal{F}$ without sharing raw graph data. Considering the potential data heterogeneity between clients, we aim to train personalized classifiers $\mathcal{F}(\theta_i)$ for each client. Formally, the training goal is to find the optimal set of parameters $\{\theta_1^*, \cdots, \theta_M^*\}$ that minimizes the average of the client losses:

$$\{\theta_1^*, \cdots, \theta_M^*\} = \operatorname{argmin} \sum_i^M \mathcal{L}_i(\mathcal{F}(\theta_i)), \ \mathcal{L}_i(\mathcal{F}(\theta_i)) = \frac{1}{n_i} \sum_j^{n_i} l(\mathcal{F}(v_j; \theta_i), y_j), \quad (1)$$

where $\mathcal{L}_i(\mathcal{F}(\theta_i))$ is the $i$-th client loss, $l(\cdot, \cdot)$ denotes the cross-entropy loss and $y_j$ is the node label.

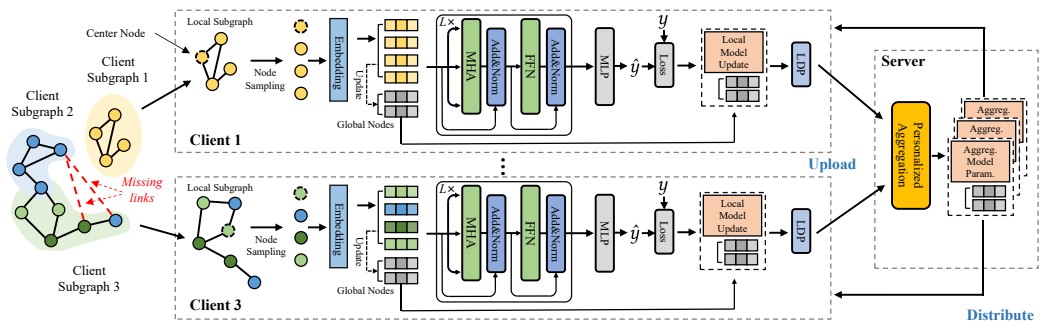

Figure 1: **The framework of FedGT**. We use a case with three clients for illustration and omit the model details of Client 2 for simplicity. The node colors indicate the node labels.

## 3.2 TRANSFORMER ARCHITECTURE

In Graph Transformer, each node is regarded as a token. Note that we also concatenate the positional encoding vectors with the input node features to supplement positional information. The Transformer architecture consists of a series of Transformer layers (Vaswani et al., 2017). Each Transformer layer has two parts: a multi-head self-attention (MHA) module and a position-wise feed-forward network (FFN). Let $\mathbf{H}_b = [\boldsymbol{h}_1, \cdots, \boldsymbol{h}_b]^\top \in \mathbb{R}^{b \times d}$ denote the input to the self-attention module where $d$ is the hidden dimension, $\boldsymbol{h}_i \in \mathbb{R}^{d \times 1}$ is the $i$-th hidden representation, and $b$ is the length of input. The MHA module firstly projects the input $\mathbf{H}_b$ to query-, key-, value-spaces, denoted as $\mathbf{Q}, \mathbf{K}, \mathbf{V}$, using three learnable matrices $\mathbf{W}_Q \in \mathbb{R}^{d \times d_K}, \mathbf{W}_K \in \mathbb{R}^{d \times d_K}$ and $\mathbf{W}_V \in \mathbb{R}^{d \times d_V}$:

$$\mathbf{Q} = \mathbf{H}_b \mathbf{W}_Q, \quad \mathbf{K} = \mathbf{H}_b \mathbf{W}_K, \quad \mathbf{V} = \mathbf{H}_b \mathbf{W}_V. \tag{2}$$

Then, in each head $i \in \{1, 2, \ldots, h\}$ ($h$ is the total number of heads), the scaled dot-product attention mechanism is applied to the corresponding $\{\mathbf{Q}_i, \mathbf{K}_i, \mathbf{V}_i\}$:

$$\text{head}_i = \text{Softmax}\left(\frac{\mathbf{Q}_i \mathbf{K}_i^T}{\sqrt{d_K}}\right) \mathbf{V}_i. \tag{3}$$

Finally, the outputs from different heads are further concatenated and transformed to obtain the final output of MHA:

$$\text{MHA}(\mathbf{H}) = \text{Concat}\left(\text{head}_1, \ldots, \text{head}_h\right) \mathbf{W}_O, \tag{4}$$

where $\mathbf{W}_O \in \mathbb{R}^{d \times d}$. In this work, we employ $d_K = d_V = d/h$ for the hidden dimensions of $\mathbf{Q}, \mathbf{K}$, and $\mathbf{V}$. Let $L$ be the total number of transformer layers and $\mathbf{H}_b^{(l)}$ be the set of input representations at the $l$-th layer. The transformer layer is formally characterized as:

$$\mathbf{H'}_b^{(l-1)} = \text{MHA}(\text{LN}(\mathbf{H}_b^{(l-1)})) + \mathbf{H}_b^{(l-1)} \tag{5}$$

$$\mathbf{H}_b^{(l)} = \text{FFN}(\text{LN}(\mathbf{H'}_b^{(l-1)})) + \mathbf{H'}_b^{(l-1)}, \ (0 \le l < L), \tag{6}$$

where layer normalizations (LN) are applied before the MHA and FFN blocks (Xiong et al., 2020). The attention in Equation. 3 brings quadratic computational complexity to the vanilla transformer.

## 4 FEDGT

In this section, we first show the scalable graph transformer with linear computational complexity to tackle the missing link issue in Sec. 4.1. Then we show the personalized aggregation to address data heterogeneity in Sec. 4.2. Furthermore, we demonstrate the local differential privacy in Sec. 4.3. Finally, we theoretically analyze the approximation error of the global attention in Sec.4.4. Figure 1 illustrates the framework of FedGT. Algorithm. 2 and 3 in Appendix. A show the pseudo codes.

## 4.1 SCALABLE GRAPH TRANSFORMER

The quadratic computational cost of the vanilla transformer inhibits its direct applications on large-scale graphs. Especially in federated learning scenarios where clients may have limited computational budgets. Here, we propose a scalable Graph Transformer with a hybrid attention scheme:

**Local Attention.** Previous works (Zhao et al., 2021; Zhang et al., 2022) show that local information is quite important for node classification. Therefore, for each center node, we randomly sample $n_s$ neighboring nodes for attention. There are several choices for the node sampling strategy such as Personalized Page Rank (PPR) (Zhang et al., 2020), GraphSage (Zhao et al., 2021), and attribute similarity (Zhang et al., 2022). We use PPR as the default sampling strategy for its superior empirical performance and explore other strategies in Appendix E.4. Specifically, the Personalized PageRank (Page et al., 1999) matrix is calculated as: $PPR = \nu(I - (1 - \nu)\overline{A}_i)^{-1}$, where factor $\nu \in [0, 1]$ (set to 0.15 in experiments), $I$ is the identity matrix and $\overline{A}_i$ denotes the column-normalized adjacency matrix of client $i$. The PPR matrix can reflect the intimacy between nodes and we use it as the sampling probability to sample nodes for local attention. It was shown in recent works (Rampášek et al., 2022; Kreuzer et al., 2021) that positional encodings (PE) are one of the most important factors in improving the performance of graph Transformers. In FedGT, we adopt the popular Laplacian positional encoding (Rampášek et al., 2022): the PE vectors are concatenated with the node features to supplement the positional information.

**Global Attention.** To preserve the advantage of the global receptive field, we propose to use a set of curated global nodes to approximate the global context of the corresponding subgraph. Section. 4.4 theoretically analyzes the approximation error. Inspired by online deep clustering (Zhan et al., 2020), we propose to dynamically update the global nodes during the training of FedGT. Algorithm 1 shows the pseudo-codes of updating global nodes with a batch of input representations

---

**Algorithm 1** Online Clustering for Global Nodes

**Input**: node batch $\mathbf{H}_b$, momentum $\gamma$, global nodes $\mu$, count of data per cluster $c$
**Output**: Updated global nodes $\mu$
1: $P = \text{FindNearest}(\mathbf{H}_b, \mu)$
2: $\mu \leftarrow c \cdot \mu \cdot \gamma + P^\top \mathbf{H}_b \cdot (1 - \gamma)$
3: $c \leftarrow c \cdot \gamma + P^\top \mathbf{1} \cdot (1 - \gamma)$
4: $\mu \leftarrow \mu / c$

---

$\mathbf{H}_b$. $\gamma$ is the hyperparameter of momentum to stabilize the updating process. FindNearest$(\cdot, \cdot)$ finds the nearest global node in Euclidean space and $P$ is the assignment matrix. The global nodes $\mu$ can be regarded as the cluster centroids of node representations and reflect the overall data distribution of the corresponding local subgraph.

Overall, during the forward pass of the graph transformer in FedGT, each node will only attend to $n_s$ sampled neighboring nodes and $n_g$ global nodes. Such a hybrid attention scheme reduces the complexity from $O(n^2)$ to $O(n(n_g + n_s))$. Since $n_g$ and $n_s$ are small constants, we obtain a linear computational complexity in FedGT.

**Comparison with GNN-based methods.** Most GNNs follow a message-passing paradigm that is likely to make false predictions with altered or missing links (Xu et al., 2019a). In contrast, the global nodes in FedGT capture the global context and get further augmented with personalized aggregation (Sec. 4.2), which can supplement the missing information. Sec. 5 shows that the hybrid attention in FedGT is robust to missing links and effectively alleviates the performance degradation.

## 4.2 PERSONALIZED AGGREGATION

Data heterogeneity is quite common in subgraph FL since local subgraphs are from different parts of the whole graph. Therefore, it is suboptimal to train a single global model for all the clients. Instead, in FedGT, we propose a personalized aggregation scheme where model parameters with similar data distribution are assigned with larger weights in the weighted averaging for each client. However, one key question is how to measure the similarity between clients under the privacy constraint. One straightforward method is to calculate the cosine similarity between local model updates. However, the similarity measured in high-dimensional parameter space is not accurate due to the curse of dimensionality (Bellman, 1966), and the large parameter size brings computational burdens. Instead, here we propose to estimate client similarity based on global nodes because global nodes can reflect the corresponding local data distribution and have limited dimension sizes. As there are no fixed orders in the global nodes, we calculate the similarity between clients $S_{i,j}$ similar to optimal transport (Villani, 2009):

$$S_{i,j} = \max_{\pi} \frac{1}{n_g} \sum_{k}^{n_g} cos(\mu_{i,k}, \mu_{j,\pi(k)}), \tag{7}$$

where $\pi$ is a bijective mapping: $[n_g] \rightarrow [n_g]$ and $cos(\cdot, \cdot)$ calculates the cosine similarity between two embedding vectors. Computing Equation. 7 is based on solving a matching problem and needs

the Hungarian algorithm with $O(n_g^3)$ complexity (Jonker & Volgenant, 1987). Note that there are also some approximation algorithms (Cuturi, 2013; Fan et al., 2017) whose complexities are $O(n_g^2)$. Since in FedGT, $n_g$ is fixed as a small constant (e.g., 10), we still use the Hungarian matching to calculate Equation. 7 which does not introduce much overhead. With the similarities, we perform weighted averaging of the updated local models as:

$$\hat{\theta}_i = \sum_j^M \alpha_{ij} \cdot \theta_j, \ \alpha_{ij} = \frac{\exp(\tau \cdot S_{i,j})}{\sum_k \exp(\tau \cdot S_{i,k})}, \tag{8}$$

where $\alpha_{ij}$ is the normalized similarity between clients $i$ and $j$, and $\hat{\theta}_i$ is the personalized weighted model parameter to send to client $i$. $\tau$ is a hyperparameter for scaling the unnormalized similarity score. Increasing $\tau$ will put more weight on parameters with higher similarities. Besides the model parameters, the global nodes from each client are also aggregated with personalized weighted averaging. Specifically, the global nodes are first aligned with optimal transport and then averaged similar to Equation. 8. In this way, the global nodes not only preserve the information of the local subgraph but can further incorporate information from other clients with similar data distributions.

### 4.3 LOCAL DIFFERENTIAL PRIVACY

Since the uploaded model parameters and the representations of global nodes may contain private information of clients, we further propose to use Local Differential Privacy (LDP) (Yang et al., 2020; Arachchige et al., 2019) to protect clients' privacy. Formally, denote the input vector of model parameters or representations as $g$, the LDP mechanism as $\mathcal{M}$, and the clipping threshold as $\delta$, we have: $\mathcal{M}(g) = clip(g, \delta) + n$, where $n \sim Laplace(0, \lambda)$ is the Laplace noise with 0 mean and noise strength $\lambda$. Previous works (Wu et al., 2021; Qi et al., 2020) show that the upper bound of the privacy budget $\epsilon$ is $\frac{2\delta}{\lambda}$, which indicates that a smaller privacy budget $\epsilon$ can be achieved by using a smaller clipping threshold $\delta$ or a larger noise strength $\lambda$. However, the classification accuracy of trained models will be negatively affected if the privacy budget is too small. Therefore, we should choose the hyperparameters of LDP appropriately to balance the model's performance and privacy protection.

### 4.4 THEORETICAL ANALYSIS OF GLOBAL ATTENTION

In FedGT, we propose to use global nodes to approximate information in the whole subgraph. Here we analyze the approximation error theoretically. For simplicity, we omit the local attention here. For ease of expression, we first define the attention score function and the output of a self-attention layer.

**Definition 1.** *Here, let $\mathbf{H} \in \mathbb{R}^{n_i \times d}$ be the node representation of the whole subgraph, where $n_i$ denotes the number of nodes in the subgraph. $\mathbf{H}_b \in \mathbb{R}^{b \times d}$ denote a batch of input nodes for self-attention. We assume the positional encodings are already fused into node representations for simplicity. $\mathbf{W}_Q, \mathbf{W}_K,$ and $\mathbf{W}_V \in \mathbb{R}^{d \times d'}$ are the weight matrices. The attention score function is:*

$$\mathcal{A}(\mathbf{H}) \triangleq \mathrm{Softmax}(\mathbf{H}_b \mathbf{W}_Q (\mathbf{H} \mathbf{W}_K)^\top). \tag{9}$$

*The output of a self-attention layer is:*

$$\mathcal{O}(\mathbf{H}) \triangleq \mathcal{A}(\mathbf{H}) \mathbf{H} \mathbf{W}_V. \tag{10}$$

**Theorem 1.** *Suppose the attention score function ($\mathcal{A}(\cdot)$) is Lipschitz continuous with constant $\mathcal{C}$. Let $\mu \in \mathbb{R}^{n_g \times d}$ denote the representations of global nodes. $P \in \mathbb{R}^{n_i \times n_g}$ is the assignment matrix to recover $\mathbf{H}$ i.e., $\mathbf{H} \approx P\mu$. Specifically, each row of $P$ is a one-hot vector indicating the global node that the node is assigned to. We assume nodes are equally distributed to the global nodes. Formally, we have the following inequality:*

$$\|\mathcal{O}(\mu) - \mathcal{O}(\mathbf{H})\|_F \le \mathcal{C} \cdot \sigma \cdot (2 + \sigma) \|\mathbf{H}\|_F^2 \cdot \|\mathbf{W}_V\|_F, \tag{11}$$

*where $\sigma \triangleq \|\mathbf{H} - P\mu\|_F / \|\mathbf{H}\|_F$ is the approximation error rate and $\| \cdot \|_F$ is the Frobenius Norm.*

*Proof.* Our idea is to bound the difference between the original and the approximate output with the approximation error rate and Lipschitz constant. Appendix C shows our detailed proof. □

Theorem. 1 indicates that we can obtain the results of self-attention with a bounded error by only computing attention with $n_g$ global nodes. Moreover, the error could be minimized if the global nodes well capture the node distributions of the whole subgraph (i.e., smaller $\sigma$).

Table 1: Node classification results of different methods in the **non-overlapping** setting. We report the means and standard deviations over three different runs (%). The best results are bolded.

| Methods | Cora | | | CiteSeer | | | Pubmed | | | All |
|---|---|---|---|---|---|---|---|---|---|---|
| | 5 Clients | 10 Clients | 20 Clients | 5 Clients | 10 Clients | 20 Clients | 5 Clients | 10 Clients | 20 Clients | Avg. |
| Local | 80.10 ± 0.76 | 77.43 ± 0.49 | 72.75 ± 0.89 | 70.10 ± 0.25 | 68.77 ± 0.35 | 64.51 ± 0.28 | 85.30 ± 0.24 | 84.88 ± 0.32 | 82.66 ± 0.65 | - |
| FedAvg | 79.63 ± 4.37 | 72.06 ± 2.18 | 69.50 ± 3.58 | 70.24 ± 0.47 | 68.32 ± 2.59 | 65.12 ± 2.15 | 84.87 ± 0.41 | 78.92 ± 0.39 | 78.21 ± 0.25 | - |
| FedPer | 81.33 ± 1.24 | 78.76 ± 0.25 | 78.24 ± 0.36 | 70.36 ± 0.34 | 70.31 ± 0.36 | 66.95 ± 0.46 | 85.88 ± 0.25 | 85.62 ± 0.23 | 84.90 ± 0.37 | - |
| GCFL+ | 80.36 ± 0.57 | 78.37 ± 0.89 | 77.19 ± 1.30 | 70.52 ± 0.64 | 69.71 ± 0.79 | 66.80 ± 0.95 | 85.77 ± 0.38 | 84.94 ± 0.35 | 84.10 ± 0.43 | - |
| FedSage+ | 80.09 ± 1.28 | 74.07 ± 1.46 | 72.68 ± 0.95 | 70.94 ± 0.21 | 69.03 ± 0.59 | 65.20 ± 0.73 | 86.03 ± 0.28 | 82.89 ± 0.37 | 79.71 ± 0.35 | - |
| FED-PUB | 83.72 ± 0.18 | 81.45 ± 0.12 | 81.10 ± 0.64 | 72.40 ± 0.26 | 71.83 ± 0.61 | 66.89 ± 0.14 | 86.81 ± 0.12 | 86.09 ± 0.17 | 84.66 ± 0.54 | - |
| Gophormer | 80.20 ± 3.66 | 74.34 ± 2.46 | 72.05 ± 3.41 | 71.22 ± 0.47 | 69.24 ± 0.60 | 65.91 ± 1.17 | 86.23 ± 0.42 | 82.31 ± 0.46 | 80.44 ± 0.57 | - |
| GraphGPS | 81.46 ± 0.70 | 75.12 ± 1.85 | 73.63 ± 4.19 | 71.19 ± 0.84 | 69.54 ± 0.70 | 65.19 ± 1.26 | 86.39 ± 0.30 | 83.40 ± 0.57 | 80.93 ± 0.52 | - |
| FedGT (Ours) | **84.41 ± 0.45** | **81.49 ± 0.41** | **81.25 ± 0.58** | **72.95 ± 0.83** | **71.98 ± 0.70** | **69.60 ± 0.45** | **87.21 ± 0.14** | **86.65 ± 0.15** | **85.79 ± 0.28** | - |

| Methods | Amazon-Computer | | | Amazon-Photo | | | ogbn-arxiv | | | All |
|---|---|---|---|---|---|---|---|---|---|---|
| | 5 Clients | 10 Clients | 20 Clients | 5 Clients | 10 Clients | 20 Clients | 5 Clients | 10 Clients | 20 Clients | Avg. |
| Local | 89.18 ± 0.15 | 88.25 ± 0.21 | 84.34 ± 0.28 | 91.85 ± 0.12 | 89.56 ± 0.09 | 85.83 ± 0.17 | 66.87 ± 0.09 | 66.03 ± 0.14 | 65.43 ± 0.21 | 78.55 |
| FedAvg | 88.03 ± 1.68 | 81.82 ± 1.71 | 78.19 ± 0.86 | 89.26 ± 1.80 | 85.31 ± 1.67 | 82.59 ± 1.18 | 66.24 ± 0.45 | 64.09 ± 0.83 | 62.47 ± 1.19 | 75.82 |
| FedPer | 88.94 ± 0.25 | 88.26 ± 0.17 | 87.85 ± 0.29 | 91.30 ± 0.33 | 89.97 ± 0.27 | 88.30 ± 0.18 | 67.02 ± 0.19 | 66.02 ± 0.27 | 65.25 ± 0.31 | 79.66 |
| GCFL+ | 89.07 ± 0.45 | 88.74 ± 0.49 | 87.81 ± 0.36 | 90.78 ± 0.69 | 90.22 ± 0.85 | 89.23 ± 1.07 | 66.97 ± 0.11 | 66.38 ± 0.14 | 65.30 ± 0.34 | 79.63 |
| FedSage+ | 89.78 ± 0.71 | 84.39 ± 1.06 | 79.75 ± 0.90 | 90.89 ± 0.44 | 86.82 ± 0.78 | 83.10 ± 0.70 | 66.91 ± 0.12 | 65.30 ± 0.13 | 62.63 ± 0.24 | 77.23 |
| FED-PUB | 90.25 ± 0.07 | 89.73 ± 0.16 | 88.20 ± 0.18 | 93.20 ± 0.15 | 92.46 ± 0.19 | 90.59 ± 0.35 | 67.62 ± 0.11 | 66.35 ± 0.16 | 63.90 ± 0.27 | 80.96 |
| Gophormer | 88.41 ± 1.21 | 83.10 ± 0.79 | 80.33 ± 0.84 | 91.34 ± 0.28 | 86.05 ± 0.51 | 83.62 ± 0.30 | 67.31 ± 0.24 | 66.32 ± 0.35 | 62.15 ± 0.32 | 77.25 |
| GraphGPS | 89.12 ± 0.23 | 84.53 ± 0.28 | 81.80 ± 0.17 | 91.45 ± 0.70 | 87.43 ± 1.06 | 83.32 ± 1.42 | 67.58 ± 0.19 | 66.15 ± 0.28 | 62.90 ± 0.33 | 77.84 |
| FedGT (Ours) | **90.78 ± 0.08** | **90.59 ± 0.09** | **90.22 ± 0.14** | **93.48 ± 0.18** | **93.17 ± 0.24** | **92.20 ± 0.16** | **68.15 ± 0.06** | **67.79 ± 0.11** | **67.53 ± 0.15** | **81.96** |

## 5 EXPERIMENTS

### 5.1 EXPERIMENTAL SETTINGS

**Datasets.** Following previous works (Zhang et al., 2021; Baek et al., 2023), we construct the distributed subgraphs from benchmark datasets by partitioning them into the number of clients, each of which has a subgraph that is part of the original graph. Specifically, we use six datasets in experiments: Cora, CiteSeer, Pubmed, and ogbn-arxiv are citation graphs (Sen et al., 2008; Hu et al., 2020b); Amazon-Computer and Amazon-Photo are product graphs (McAuley et al., 2015; Shchur et al., 2018). We use the METIS (Karypis & Kumar, 1995) as the default graph partitioning algorithm, which can specify the number of subgraphs. Specifically, METIS firstly coarsens the original graph into coarsened graphs with maximal matching methods (Karypis & Kumar, 1998). It then computes a minimum edge-cut partition on the coarsened graph. Finally, the subgraphs are obtained by projecting the partitioned coarsened graph back to the original graph. We also consider Louvain partitioning (Blondel et al., 2008) in Appendix E.

Inspired by real-world applications, we further consider two subgraph settings. In the **1) non-overlapping setting**, there is no overlapped nodes between subgraphs. For example, for a period, the local resident-location relationship graph is only stored in one city's database. We directly use the outputs from the METIS since it provides the non-overlapping partitioned subgraphs. We consider 5/10/20 clients (subgraphs) in this setting. In the **2) overlapping setting**, there are overlapped nodes between subgraphs. For example, the same custormer may have multiple accounts at different banks. We randomly sample the subgraphs multiple times from the partitioned graph. Specifically, we first divide the original graph into 2/6/10 disjoint subgraphs with METIS for 10/30/50 clients settings. After that, for each partitioned subgraph from METIS, we randomly sample half of the nodes and the associated edges as one subgraph for 5 times. Therefore, the generated subgraphs belonging to the same METIS partition have overlapped nodes and form cluster structures. Note that due to the different subgraph sampling manners, the numbers of clients are different in the two settings.

For dataset splitting, 20%/40%/40% nodes from each subgraph are randomly sampled for training, validation, and testing except for the ogbn-arxiv dataset. This is because the ogbn-arxiv dataset has a relatively larger number of nodes as shown in Table .3. Therefore, we randomly sample 5% nodes for training, the remaining half of the nodes for validation, and the other nodes for testing.

**Baselines.** FedGT is compared with popular FL frameworks (FedAvg (McMahan et al., 2017b), FedPer (Arivazhagan et al., 2019)), FGL methods (GCFL+ (Xie et al., 2021a), FedSage+ (Zhang et al., 2021), FED-PUB (Baek et al., 2023)), and graph transformers (Gophormer (Zhao et al., 2021), GraphGPS (Rampášek et al., 2022)) extended to the subgraph FL setting for comprehensive evaluations. We also train FedGT locally without sharing model parameters for reference (Local). More baseline details are in Appendix. D.

**Implementation Details.** For the classifier model in FedAvg, FedPer, GCFL+, and FedSage+, we use GraphSage (Hamilton et al., 2017b) with two layers and the mean aggregator following previous

Table 2: Node classification results of different methods in the **overlapping** setting. We report the means and standard deviations over three different runs (%). The best results are bolded.

| Methods | Cora | | | CiteSeer | | | Pubmed | | | All |
| | 10 Clients | 30 Clients | 50 Clients | 10 Clients | 30 Clients | 50 Clients | 10 Clients | 30 Clients | 50 Clients | Avg. |
|---|---|---|---|---|---|---|---|---|---|---|
| Local | $78.14 \pm 0.15$ | $73.60 \pm 0.18$ | $69.87 \pm 0.40$ | $68.94 \pm 0.29$ | $66.13 \pm 0.49$ | $63.70 \pm 0.92$ | $84.90 \pm 0.05$ | $83.27 \pm 0.24$ | $80.88 \pm 0.19$ | - |
| FedAvg | $78.55 \pm 0.49$ | $69.56 \pm 0.79$ | $65.19 \pm 3.88$ | $68.73 \pm 0.46$ | $65.02 \pm 0.59$ | $63.85 \pm 1.31$ | $84.66 \pm 0.11$ | $80.62 \pm 0.46$ | $80.18 \pm 0.50$ | - |
| FedPer | $78.84 \pm 0.32$ | $73.46 \pm 0.43$ | $72.54 \pm 0.52$ | $70.42 \pm 0.26$ | $65.09 \pm 0.48$ | $64.04 \pm 0.46$ | $85.76 \pm 0.14$ | $83.45 \pm 0.15$ | $81.90 \pm 0.23$ | - |
| GCFL+ | $78.60 \pm 0.25$ | $73.41 \pm 0.36$ | $73.13 \pm 0.87$ | $69.80 \pm 0.34$ | $65.17 \pm 0.32$ | $64.71 \pm 0.67$ | $85.08 \pm 0.21$ | $83.77 \pm 0.17$ | $80.95 \pm 0.22$ | - |
| FedSage+ | $79.01 \pm 0.30$ | $72.20 \pm 0.76$ | $66.52 \pm 1.37$ | $70.09 \pm 0.26$ | $66.71 \pm 0.18$ | $64.89 \pm 0.25$ | $86.07 \pm 0.06$ | $83.26 \pm 0.08$ | $80.48 \pm 0.20$ | - |
| FED-PUB | $79.65 \pm 0.17$ | $75.42 \pm 0.48$ | $73.13 \pm 0.29$ | $70.43 \pm 0.27$ | $67.41 \pm 0.36$ | $65.13 \pm 0.40$ | $85.60 \pm 0.10$ | $85.19 \pm 0.15$ | $84.26 \pm 0.19$ | - |
| Gophormer | $78.74 \pm 0.42$ | $73.41 \pm 0.77$ | $68.30 \pm 0.46$ | $69.53 \pm 0.21$ | $65.89 \pm 0.45$ | $63.15 \pm 0.63$ | $85.47 \pm 0.06$ | $82.14 \pm 0.25$ | $80.85 \pm 0.13$ | - |
| GraphGPS | $79.40 \pm 0.46$ | $75.42 \pm 0.75$ | $69.07 \pm 2.16$ | $69.95 \pm 0.30$ | $65.77 \pm 0.47$ | $62.54 \pm 0.59$ | $85.49 \pm 0.17$ | $82.73 \pm 0.16$ | $80.50 \pm 0.33$ | - |
| FedGT (Ours) | $\mathbf{81.73} \pm 0.26$ | $\mathbf{77.94} \pm 0.56$ | $\mathbf{76.20} \pm 0.39$ | $\mathbf{72.40} \pm 0.45$ | $\mathbf{68.86} \pm 0.33$ | $\mathbf{67.71} \pm 0.67$ | $\mathbf{86.90} \pm 0.14$ | $\mathbf{86.15} \pm 0.18$ | $\mathbf{84.95} \pm 0.17$ | - |

| Methods | Amazon-Computer | | | Amazon-Photo | | | ogbn-arxiv | | | All |
| | 10 Clients | 30 Clients | 50 Clients | 10 Clients | 30 Clients | 50 Clients | 10 Clients | 30 Clients | 50 Clients | Avg. |
|---|---|---|---|---|---|---|---|---|---|---|
| Local | $89.17 \pm 0.09$ | $85.77 \pm 0.13$ | $81.40 \pm 0.05$ | $91.65 \pm 0.15$ | $86.20 \pm 0.16$ | $84.71 \pm 0.09$ | $66.95 \pm 0.05$ | $64.62 \pm 0.07$ | $62.89 \pm 0.06$ | 76.82 |
| FedAvg | $87.65 \pm 0.28$ | $77.56 \pm 0.71$ | $76.41 \pm 0.95$ | $89.90 \pm 0.06$ | $81.42 \pm 0.15$ | $76.98 \pm 0.68$ | $66.86 \pm 0.04$ | $62.15 \pm 0.12$ | $60.81 \pm 0.27$ | 74.23 |
| FedPer | $89.52 \pm 0.05$ | $86.79 \pm 0.16$ | $85.64 \pm 0.19$ | $90.23 \pm 0.24$ | $90.05 \pm 0.19$ | $88.37 \pm 0.18$ | $67.21 \pm 0.08$ | $65.00 \pm 0.37$ | $62.19 \pm 0.46$ | 77.76 |
| GCFL+ | $88.65 \pm 0.13$ | $86.52 \pm 0.12$ | $84.30 \pm 0.25$ | $91.42 \pm 0.14$ | $90.12 \pm 0.15$ | $88.67 \pm 0.11$ | $67.10 \pm 0.08$ | $64.33 \pm 0.17$ | $62.87 \pm 0.10$ | 77.70 |
| FedSage+ | $88.61 \pm 0.18$ | $80.24 \pm 0.30$ | $78.92 \pm 0.27$ | $90.26 \pm 0.45$ | $82.57 \pm 0.34$ | $78.52 \pm 0.20$ | $67.38 \pm 0.13$ | $64.89 \pm 0.09$ | $62.28 \pm 0.14$ | 75.72 |
| FED-PUB | $89.94 \pm 0.09$ | $89.10 \pm 0.07$ | $88.34 \pm 0.15$ | $92.78 \pm 0.06$ | $91.14 \pm 0.09$ | $90.45 \pm 0.17$ | $64.20 \pm 0.08$ | $62.97 \pm 0.14$ | $61.85 \pm 0.15$ | 78.72 |
| Gophormer | $89.03 \pm 0.10$ | $82.89 \pm 0.48$ | $80.45 \pm 0.76$ | $91.74 \pm 0.08$ | $84.20 \pm 0.74$ | $79.16 \pm 1.56$ | $67.42 \pm 0.06$ | $64.98 \pm 0.30$ | $62.55 \pm 0.23$ | 76.11 |
| GraphGPS | $89.55 \pm 0.21$ | $83.97 \pm 0.26$ | $81.06 \pm 0.47$ | $91.97 \pm 0.10$ | $84.11 \pm 0.27$ | $81.59 \pm 1.85$ | $67.27 \pm 0.21$ | $63.96 \pm 0.09$ | $62.71 \pm 0.15$ | 76.50 |
| FedGT (Ours) | $\mathbf{90.76} \pm 0.10$ | $\mathbf{89.98} \pm 0.15$ | $\mathbf{89.04} \pm 0.12$ | $\mathbf{93.19} \pm 0.15$ | $\mathbf{92.03} \pm 0.14$ | $\mathbf{91.37} \pm 0.20$ | $\mathbf{68.78} \pm 0.13$ | $\mathbf{67.92} \pm 0.11$ | $\mathbf{65.78} \pm 0.26$ | 80.63 |

work (Zhang et al., 2021). The number of nodes sampled in each layer of GraphSage is 5. The first layer is regarded as the base layer in FedPer. For FED-PUB, we use two layers of GCN (Kipf & Welling, 2017) following the original paper (Baek et al., 2023). We use Gophormer and GraphGPS with 2 layers and 4 attention heads as their backbone models and extend them to FL with the FedAvg framework. More implementation details of FedGT and FL training are shown in Appendix.D.

## 5.2 EXPERIMENTAL RESULTS.

**Main Results.** We show the average node classification results in non-overlapping and overlapping settings in Table 1 and 2. Generally, FedGT consistently overperforms all the baseline methods in both settings. Specifically, we have the following observations and insights: **1)** All model performance deteriorates when the number of clients increases. This is because the size of local data decreases and the number of missing links increases (see Appendix E.5) with a larger number of clients. The data heterogeneity issue also becomes more severe (see Appendix E.6). It is therefore more challenging to train local models and collaboratively learn generalizable models with other clients. **2)** Generally, the non-overlapping setting is more challenging than overlapping with the same number of clients. This is mainly because subgraphs in the non-overlapping setting are completely disjoint and more heterogeneous. Moreover, the non-overlapping setting has less number of nodes due to the experimental designs. **3)** The methods based on graph transformers (i.e., Gophormer and GraphGPS) are indeed powerful and can achieve competitive node classification results in FGL. For instance, GraphGPS achieves 81.46 % on the Cora dataset, and with 5 clients (the highest accuracy besides FedGT). However, they all fail to consider the heterogeneity of clients and can hardly work in settings with a larger number of clients (e.g., GraphGPS drops to 73.63 % with 20 clients). The same circumstance occurs to FedSage+ and FedAvg. **4)** FedGT leverages the advantage of a hybrid graph attention framework and personalized aggregation and can achieve consistent improvement over baselines. For example, the classification accuracy of FedGT only decreases from 90.78% to 90.22% when the number of clients increases from 5 to 20 on Amazon-Computer (the number of missing links increases by 54,878 and the data heterogeneity increases to 0.759), indicating its robustness to missing links and data heterogeneity. Moreover, FedGT also overperforms its variant that only trains models locally (Local), which shows the ability of FedGT to effectively leverage distributed subgraph data for joint performance improvement. Finally, Figure 5 and 6 in the Appendix E.1 show the convergence plots. FedGT can converge more rapidly than the other baselines.

**Effectiveness of Similarity Estimation.** Here we show whether FedGT can accurately identify clients with similar data distributions in personalized aggregation. We regard two clients with similar data distributions if they have similar label distributions, i.e., higher cosine similarity between label distribution vectors. In Figure 2(a), we show the heatmaps of pairwise label distribution similarity (cosine similarity) over 20 clients in the overlapping setting. There are clearly four clusters by the interval of five in the diagonal and the last two form a larger cluster. In Figure 2(b) and (c), we show the normalized similarity (i.e., $\alpha_{ij}$ in Equation. 8) of FedGT and FedGT w/o optimal transport where we directly calculate cosine similarity without aligning. We observe that Figure 2(b) can well capture the four clusters and identify the less-obvious larger cluster, verifying the effectiveness of our

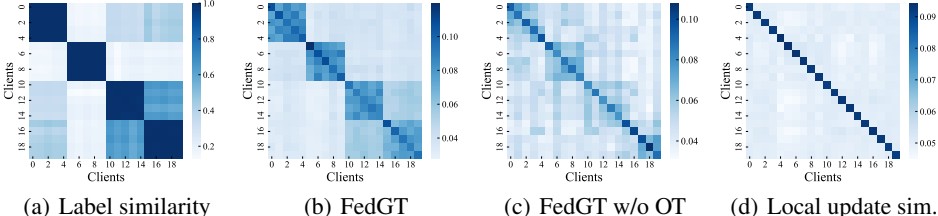

| (a) Label similarity | (b) FedGT | (c) FedGT w/o OT | (d) Local update sim. |

Figure 2: Similarity Heatmaps in the overlapping setting on Cora. (a) measures the cosine similarity of label distributions. (b) and (c) shows the normalized similarity in FedGT and FedGT w/o optimal transport (OT); (d) shows the normalized cosine similarity of local model updates at round 30.

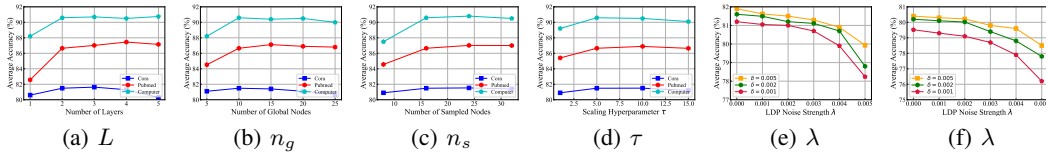

| (a) $L$ | (b) $n_g$ | (c) $n_s$ | (d) $\tau$ | (e) $\lambda$ | (f) $\lambda$ |

Figure 3: Hyperparameter analysis in the non-overlapping setting (10 clients) on Cora. (a), (b), (c), and (d) show the influence of the number of layers $L$, global nodes $n_g$, sampled nodes $n_s$, and scaling hyperparameter $\tau$. (e) and (f) explore the influence of $\delta$ and $\lambda$ in LDP. We apply LDP only to the uploaded global nodes (e) or both the global nodes and local model updates (f).

similarity estimation scheme based on the global nodes. In contrast, the cluster structure in Figure 2(c) becomes blurred and noisy, showing the necessity of optimal transport for aligning global nodes. In Figure 2(d), we further show the normalized cosine similarity of local model updates, which can not reflect any cluster structures and each client only has high similarity to itself. We conjecture it is due to the high dimension of the parameter space and the randomness of stochastic gradient descent. Moreover, using label distribution similarity may lead to additional privacy risks (see Appendix F).

**Hyperparameter Analysis.** We vary several important hyperparameters in Figure 3 to show their influence on FedGT. Generally, FedGT is robust to the choice of hyperparameters. In Figure 3(a), we observe that the performance increases at the beginning with the increase of $L$ due to the improved model capability. However, the performance slightly decreases when $L$ becomes 4 or 5, possibly suffering from over-fitting. Figure 3(b) and (c) show the influence of $n_g$ and $n_s$. Generally, FedGT has better performance with larger $n_g$ and $n_s$. Figure 3(d) indicates that choosing an appropriate $\tau$ can balance the weight of local model and other clients' models and help improve the performance.

**Trade-off between Privacy and Accuracy.** In Figure 3(e) and (f), we explore the influence of $\delta$ and $\lambda$ in LDP. The influence on privacy budget $\epsilon$ is shown in Appendix E.8. Generally, larger $\lambda$ values and smaller $\delta$ values result in better privacy protection, while also leading to a larger performance decline, especially when LDP is applied on both local model updates and global nodes. Therefore, we need to select $\lambda$ and $\delta$ appropriately to achieve a balance between privacy protection and model performance.

**Ablation Studies and Complexity Analysis.** The ablations studies demonstrating the effectiveness of proposed modules in FedGT and more complexity analysis are shown in Appendix.E.

# 6 CONCLUSION

To tackle the challenges in subgraph FL (missing links and data heterogeneity), a novel scalable Federated Graph Transformer (FedGT) is proposed in this paper. The linear-complexity hybrid attention scheme enables FedGT to capture both local and global information and be robust to missing links. Personalized aggregation based on the similarity of global nodes from different clients is leveraged to deal with data heterogeneity. Local differential privacy is further used to protect data privacy. Finally, experiments on various subgraph FL settings show the superiority of FedGT. We discuss the limitations and future works in the Appendix. H.

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

## A  ALGORITHMS OF FEDGT

In Algorithm. 2 and 3, we show the pseudo-codes for the clients and server in FedGT.

---

**Algorithm 2 FedGT** Client Algorithm

---

1: **Input**: number of local epochs $E$, learning rate $\eta$, local model $\mathcal{F}_i$, loss function $\mathcal{L}_i$
2: **Function** InitClient()
3: Calculate PPR matrix and Positional Encoding.

4: **Function** RunClient($\hat{\theta}_i, \hat{\mu}_i$)
5: $\theta_i \leftarrow \hat{\theta}_i, \mu_i \leftarrow \hat{\mu}_i$.
6: **for** each local epoch $e$ from 1 to $E$ **do**
7:     $\theta_i \leftarrow \theta_i - \eta \nabla \mathcal{L}_i(\mathcal{F}_i(\theta_i))$.
8:     Update global node $\mu_i$ with Algorithm. 1.
9: **end for**
10: $\Delta \theta_i = \theta_i - \hat{\theta}_i$; apply LDP to $\Delta \theta_i, \mu_i$.
11: **return** $\Delta \theta_i, \mu_i$

---

**Algorithm 3 FedGT** Server Algorithm

---

1: **Input**: Total rounds $R$, number of clients $M$, scaling factor $\tau$.
2: **Function** RunServer()
3: initialize $\hat{\theta}^{(1)}$ and $\hat{\mu}^{(1)}$
4: **for** $r = 1, 2, \cdots, R$ **do**
5:     **for** client $i \in \{1, 2, \cdots, M\}$ **in parallel do**
6:         **if** $r = 1$ **then**
7:             InitClient().
8:             $\Delta \theta_i, \mu_i^{(r+1)} \leftarrow$ RunClient($\hat{\theta}^{(r)}, \hat{\mu}^{(r)}$).
9:             $\theta_i^{(r)} \leftarrow \hat{\theta}^{(r)} + \Delta \theta_i$.
10:        **else**
11:            Calculate $S_{i,j}$ with Equation. 7.
12:            $\hat{\theta}_i^{(r)} \leftarrow \sum_{j=0}^{M} \frac{\exp(\tau \cdot S_{i,j})}{\sum_{k=0}^{M} \exp(\tau \cdot S_{i,k})} \theta_j^{(r)}$.
13:            Obtain $\hat{\mu}_i^{(r)}$ with aligning and weighted averaging.
14:            $\Delta \theta_i, \mu_i^{(r+1)} \leftarrow$ RunClient($\hat{\theta}_i^{(r)}, \hat{\mu}_i^{(r)}$).
15:            $\theta_i^{(r+1)} \leftarrow \hat{\theta}_i^{(r+1)} + \Delta \theta_i$
16:        **end if**
17:    **end for**
18: **end for**

---

## B  MORE DISCUSSIONS OF CHALLENGES IN SUBGRAPH FL

Subgraph federated learning has brought unique challenges. Firstly, different from federated learning in other domains such as CV and NLP, where data samples of images and texts are isolated and independent, nodes in graphs are connected and correlated. Therefore, there are potentially missing links/edges between subgraphs that are not captured by any client, leading to information loss or bias. This becomes even worse when GNNs are used as graph mining models because most GNNs follow a message-passing scheme that aggregates node embeddings along edges (Gilmer et al., 2017; Hamilton et al., 2017a). For example, Figure 4(a) shows an illustration of missing links between subgraphs. Figure 4(b) shows the constructed rooted tree of the example center node by a 2-layer GNN on the local subgraph and the global graph. In the message-passing process, the node embeddings are aggregated bottom-to-up and finally the embedding of the center node is used for its node label prediction. Due to the missing link issue, the rooted tree on the local graph is biased and GNN is prone to make a false prediction (dark green instead of blue) in such a situation.

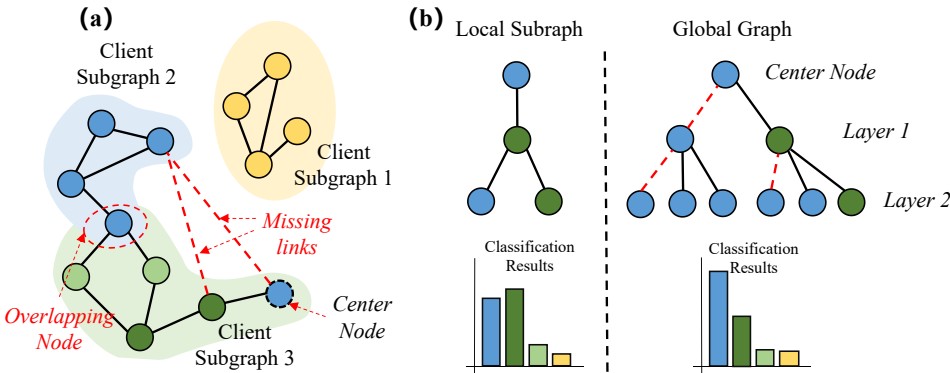

Figure 4: **(a) An illustration of subgraph federated Learning.** There are three clients (subgraphs) and the color of each node indicates its label. Two main challenges are missing links and data heterogeneity. **(b) The rooted tree on the local subgraph is biased** due to the missing links, and the GNN is prone to make a false prediction based on the local subgraph.

## C   PROOF OF THEOREM 1

**Theorem 1.** *Suppose the attention score function $(\mathcal{A}(\cdot))$ is Lipschitz continuous with constant $\mathcal{C}$. Let $\mu \in \mathbb{R}^{n_g \times d}$ denote the representations of global nodes. $P \in \mathbb{R}^{n_i \times n_g}$ is the assignment matrix to recover $\mathbf{H}$ i.e., $\mathbf{H} \approx P\mu$. Specifically, each row of $P$ is a one-hot vector indicating the global node that the node is assigned to. We assume nodes are equally distributed to the global nodes. Formally, we have the following inequality:*

$$\|\mathcal{O}(\mu) - \mathcal{O}(\mathbf{H})\|_F \leq \mathcal{C} \cdot \sigma \cdot (2 + \sigma)\|\mathbf{H}\|_F^2 \cdot \|\mathbf{W}_V\|_F, \tag{12}$$

*where $\sigma \triangleq \|\mathbf{H} - P\mu\|_F/\|\mathbf{H}\|_F$ is the approximation error rate and $\|\cdot\|_F$ is the Frobenius Norm of matrices.*

*Proof.* We first show that $\mathcal{O}(P\mu) = \mathcal{O}(\mu)$ as below:

$$\mathcal{O}(P\mu) = \mathcal{A}(P\mu)P\mu\mathbf{W}_V \tag{13}$$

$$= \text{Softmax}(\mathbf{H}_b\mathbf{W}_Q(P\mu\mathbf{W}_K)^\top)P\mu\mathbf{W}_V \tag{14}$$

$$= \text{Softmax}(\mathbf{H}_b\mathbf{W}_Q(\mu\mathbf{W}_K)^\top + \log(\mathbf{1}_{n_i}P))\mu\mathbf{W}_V \tag{15}$$

$$= \text{Softmax}(\mathbf{H}_b\mathbf{W}_Q(\mu\mathbf{W}_K)^\top)\mu\mathbf{W}_V \tag{16}$$

$$= \mathcal{O}(\mu) \tag{17}$$

where the first two equations are based on the definitions of $\mathcal{A}(\cdot)$ and $\mathcal{O}(\cdot)$. The third equation is based on the calculation mechanism of Softmax. The fourth equation uses the assumption that each global node is assigned with the same number of nodes i.e., $\mathbf{1}_{n_i}P = n_i/n_g \cdot \mathbf{1}_{n_g}$. Here, $\mathbf{1}$ denotes a vector with 1 in each entry and the specific subscript denotes its dimension. Moreover, the above equation holds because the self-attention layer is, by definition, a permutation-invariant operator. Then we have:

$$\|\mathcal{O}(\mu) - \mathcal{O}(\mathbf{H})\|_F = \|\mathcal{O}(P\mu) - \mathcal{O}(\mathbf{H})\|_F \tag{18}$$

$$= \|\mathcal{A}(P\mu)P\mu\mathbf{W}_V - \mathcal{A}(\mathbf{H})\mathbf{H}\mathbf{W}_V\|_F \tag{19}$$

$$\leq \|\mathcal{A}(P\mu)P\mu - \mathcal{A}(\mathbf{H})\mathbf{H}\|_F\|\mathbf{W}_V\|_F \tag{20}$$

For the left term of the last line, with triangle inequality, we have:

$$\|\mathcal{A}(P\mu)P\mu - \mathcal{A}(\mathbf{H})\mathbf{H}\|_F \tag{21}$$

$$= \|\mathcal{A}(P\mu)P\mu - \mathcal{A}(\mathbf{H})P\mu + \mathcal{A}(\mathbf{H})P\mu - \mathcal{A}(\mathbf{H})\mathbf{H}\|_F \tag{22}$$

$$\leq \|\mathcal{A}(P\mu)P\mu - \mathcal{A}(\mathbf{H})P\mu\|_F + \|\mathcal{A}(\mathbf{H})P\mu - \mathcal{A}(\mathbf{H})\mathbf{H}\|_F \tag{23}$$

$$\leq \|\mathcal{A}(P\mu) - \mathcal{A}(\mathbf{H})\|_F\|P\mu\|_F + \|\mathcal{A}(\mathbf{H})\|_F\|P\mu - \mathbf{H}\|_F \tag{24}$$

Based on the assumption that the attention score function $(\mathcal{A}(\cdot))$ is Lipschitz continuous with constant $\mathcal{C}$, we have:

$$\|\mathcal{A}(P\mu) - \mathcal{A}(\mathbf{H})\|_F \leq \mathcal{C}\|P\mu - \mathbf{H}\|_F \leq \mathcal{C} \cdot \sigma\|\mathbf{H}\|_F. \tag{25}$$

Moreover, we have:

$$\|P\mu\|_F = \|P\mu - \mathbf{H} + \mathbf{H}\|_F \leq \|P\mu - \mathbf{H}\|_F + \|\mathbf{H}\|_F \leq (1+\sigma)\|\mathbf{H}\|_F, \tag{26}$$

which is based on the triangle inequality and the definition of approximation error rate. $\|\mathcal{A}(\mathbf{H})\|_F$ can also be bounded by $\mathcal{C}\|\mathbf{H}\|_F$ with the Lipschitz continuity assumption. Finally, we have:

$$\|\mathcal{O}(\mu) - \mathcal{O}(\mathbf{H})\|_F \tag{27}$$

$$\leq (\|\mathcal{A}(P\mu) - \mathcal{A}(\mathbf{H})\|_F \|P\mu\|_F + \|\mathcal{A}(\mathbf{H})\|_F \|P\mu - \mathbf{H}\|_F)\|\mathbf{W}_V\|_F \tag{28}$$

$$\leq \mathcal{C} \cdot \sigma \cdot (2+\sigma)\|\mathbf{H}\|_F^2 \cdot \|\mathbf{W}_V\|_F. \tag{29}$$

$$\square$$

# D MORE DETAILS OF EXPERIMENTAL SETTINGS

## D.1 DATASET STATISTICS

Table 3: Dataset statistics

| Datasets | Nodes | Edges | Classes | Features |
|---|---|---|---|---|
| Cora | 2,708 | 5,429 | 7 | 1,433 |
| Citeseer | 3,327 | 4,732 | 6 | 3,703 |
| PubMed | 19,717 | 44,324 | 3 | 500 |
| Amazon-Computer | 13,752 | 491,722 | 10 | 767 |
| Amazon-Photo | 7,650 | 238,162 | 8 | 745 |
| ogbn-arxiv | 169,343 | 2,315,598 | 40 | 128 |

## D.2 BASELINES

- **FedAvg** (McMahan et al., 2017b) aggregates the uploaded models with respect to the number of training samples.

- **FedPer** (Arivazhagan et al., 2019) is a personalized FL baseline, which only shares the base layers while training personalized classification layers locally for each client.

- **GCFL+** (Xie et al., 2021a) is a graph-level clustered FL framework to deal with data heterogeneity. tasks. Specifically, it iteratively bi-partitions clients into two disjoint client groups based on their gradient similarities. Then, the model weights are only shared between grouped clients having similar gradients to deal with data heterogeneity. Note that the clustering scheme in our FedGT is quite different from GCFL+: In FedGT, we use online clustering to update the representations of global nodes in each client; the server of GCFL+ uses bi-partitioning to cluster clients. GCFL+ was originally designed for graph classification and we adapt it to our setting here.

- **FedSage+** (Zhang et al., 2021) trains a missing neighbor generator along with the classifier to deal with missing links. Each client first receives node representations from other clients and then calculates the corresponding gradients. The gradients are sent back to other clients to train the graph generators.

- **FED-PUB**(Baek et al., 2023) is personalized subgraph-level FL baseline. FED-PUB utilizes functional embeddings of the local GNNs using random graphs as inputs to compute similarities between them and use the similarities to perform weighted averaging for server-side aggregation. Furthermore, it learns a personalized sparse mask at each client to select and update only the subgraph-relevant subset of the aggregated parameters.

- **Gophormer** (Zhao et al., 2021) is a scalable graph transformer that samples nodes for attention with GraphSage(Hamilton et al., 2017b). We extend it to the subgraph FL setting with FedAvg.

- **GraphGPS** (Rampášek et al., 2022) is a powerful graph transformer framework. We use GIN(Xu et al., 2019b), Performer (Choromanski et al., 2021), and Laplacian positional encoding as the corresponding modules. We extend it to the subgraph FL setting with FedAvg.

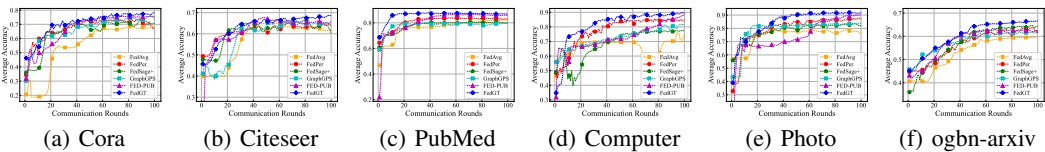

(a) Cora     (b) Citeseer     (c) PubMed     (d) Computer     (e) Photo     (f) ogbn-arxiv

Figure 5: The average testing accuracy in the non-overlapping setting with 10 clients over 100 rounds.

(a) Cora     (b) Citeseer     (c) PubMed     (d) Computer     (e) Photo     (f) ogbn-arxiv

Figure 6: The average testing accuracy in the overlapping setting with 30 clients over 100 rounds.

## D.3 IMPLEMENTATION DETAILS

For the classifier model in FedAvg, FedPer, GCFL+, and FedSage+, we use GraphSage (Hamilton et al., 2017b) with two layers and the mean aggregator following previous work (Zhang et al., 2021). The number of nodes sampled in each layer of GraphSage is 5. The first layer is regarded as the base layer in FedPer. For FED-PUB, we use two layers of Graph Convolutional Network (GCN) (Kipf & Welling, 2017) following the original paper (Baek et al., 2023). We use Gophormer and GraphGPS with 2 layers and 4 attention heads as their backbone models and extend them to FL with the FedAvg framework. For the other hyperparameter settings of baselines, we use the default settings in their original papers. The code of FedGT will be made public upon paper acceptance.

In our FedGT, we set the scaling hyperparameter $\tau$ as 5. $n_s$ and $n_g$ are set as 16 and 10 respectively. The momentum hyperparameter $\gamma$ in Algorithm. 1 is set as 0.9. The number of layers $L$ is set as 2 and the number of attention heads is set as 4. The dimension of Laplacian position encoding is 8 in the default setting. Since LDP is not applied to the local model updates in all the other baselines. In the default setting, we apply LDP with $\delta = 0.002$ and $\lambda = 0.001$ only to the uploaded global nodes for a fair comparison.

In FL training, considering the dataset size, we perform 100 communication rounds with 1 local training epoch for Cora, Citeseer, and PubMed. We set the number of communication rounds and local training epochs as 200 and 2 respectively for the other datasets. For all models, the hidden dimension is 128. We use a batch size of 64, and an Adam (Kingma & Ba, 2014) optimizer with a learning rate of 0.001 and weight decay $5e-4$ for local training. All clients participate in the FL training in each round. For evaluation, we calculate the node classification accuracy on subgraphs at the client side and then average all clients over three different runs with random seeds. The test accuracy for all the models is reported at their best validation epoch. We implement all experiments with Python (3.9.13) and Pytorch (1.12.1) on an NVIDIA Tesla V100 GPU.

## E MORE EXPERIMENTAL RESULTS

Here we show more experimental results and data statistics.

### E.1 TESTING ACCURACY CURVES

Figure 5 and 6 show the convergence plots. FedGT can converge more rapidly than the other baselines.

### E.2 MORE PARTITIONING METHODS

Besides the METIS graph partitioning algorithm used in the main text, we also try another popular graph partitioning method Louvain (Blondel et al., 2008) to evaluate FedGT (Table 4). Following

(Zhang et al., 2021), we find hierarchical graph clusters on each dataset with the Louvain algorithm (Blondel et al., 2008) and use the clustering results with 10 clusters of similar sizes to obtain subgraphs for clients. The training/validation/testing ratio is 60%/20%/20% according to (Zhang et al., 2021). In Table 4, we observe that our FedGT can also outperform baseline methods in the Louvain partitioning setting, showing its effectiveness and generalizability.

Table 4: Node classification results with the Louvain graph partitioning algorithms and 10 clients. We report the means and standard deviations over three different runs (%). The best results are bolded.

| Model | Cora | CiteSeer | PubMed |
|---|---|---|---|
| FedAvg | 86.15±0.60 | 71.24±0.35 | 86.87±0.12 |
| FedPer | 86.73±0.22 | 73.90±0.62 | 87.51±0.11 |
| GCFL+ | 86.45±0.37 | 73.84±0.24 | 87.60±0.34 |
| FedSage+ | 86.34±0.42 | 73.95±0.29 | 87.45±0.18 |
| FED-PUB | 86.78±0.20 | 74.03±0.31 | 87.40±0.26 |
| Gophormer | 86.02±0.28 | 72.76±0.40 | 86.92±0.43 |
| GraphGPS | 86.10±0.53 | 72.55±0.68 | 86.57±0.21 |
| FedGT | **87.29**±0.25 | **74.21**±0.37 | **87.95**±0.09 |

### E.3 COMPUTATIONAL AND COMMUNICATION COST.

In Table 5, we compare the computational and communication cost of FedGT with baselines. The computational cost records the average time to finish an FL round and the communication cost measures the total size of transmitted parameters between clients and the server in a round. Overall, the computational and communication overhead of FedGT is acceptable, considering its consistent performance improvement compared with baseline methods. The computational and communication cost of FedGT can be further optimized with model compression and acceleration methods such as parameter quantization (Deng et al., 2020), which we left for future exploration.

Table 5: Communication and Computational cost comparisons with baselines. Non-overlapping setting with 5 clients on the Cora dataset is used here.

| Model | Communication Cost (%) | Computational Cost (%) |
|---|---|---|
| FedAvg | 100.0 | 100.0 |
| FedPer | 86.5 | 96.2 |
| GCFL+ | 100.0 | 107.4 |
| FedSage+ | 284.8 | 162.9 |
| FED-PUB | 89.6 | 127.8 |
| Gophormer | 152.9 | 118.7 |
| GraphGPS | 231.5 | 146.1 |
| FedGT | 154.1 | 126.3 |

### E.4 INFLUENCE OF NODE SAMPLING STRATEGIES

Figure 7 compares three node sampling strategies for FedGT in the non-overlapping setting. The sampling depth in GraphSAGE is set to 2 and we calculate the cosine similarity of node features in Attribute Similarity. For each sampling strategy, we sample the same number of nodes ($n_s = 16$) for a fair comparison. Specifically, $n_s$ nodes are randomly sampled in GraphSAGE's sampled node pool and the calculated Attribute Similarity is regarded as the sampling probability to sample $n_s$ nodes. We can observe that FedGT is robust to the choice of sampling strategies and PPR has an advantage over the other two strategies. Therefore, we use PPR to sample nodes for local attention in FedGT in the default setting.

### E.5 NUMBER OF MISSING LINKS

In Table 6 and 7, we show the total number of missing links in the non-overlapping and overlapping settings with different numbers of clients. We can observe that there are more missing links with

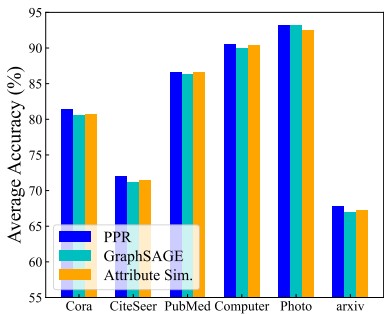

Figure 7: Influence of node sampling strategies.

the increased number of clients, which leads to more severe information loss in subgraph federated learning. We also note that the overlapping setting has more missing links than the non-overlapping setting with the same number of clients, e.g., 10 clients. This is mainly due to the node sampling procedure in the overlapping setting. Some links in the partitioned graph are not sampled and included in any client.

Table 6: Total number of missing links with different numbers of clients (non-overlapping).

| Dataset | 5 clients | 10 clients | 20 clients |
|---|---|---|---|
| Cora | 403 | 615 | 853 |
| CiteSeer | 105 | 304 | 424 |
| PubMed | 3,388 | 5,969 | 8,254 |
| Amazon-Computer | 34,578 | 65,098 | 89,456 |
| Amazon-Photo | 28,928 | 22,434 | 33,572 |
| ogbn-arxiv | 130,428 | 246,669 | 290,249 |

Table 7: Total number of missing links with respect to the number of clients (overlapping).

| Dataset | 10 clients | 30 clients | 50 clients |
|---|---|---|---|
| Cora | 1,391 | 1,567 | 1,733 |
| CiteSeer | 922 | 1,043 | 1,160 |
| PubMed | 11,890 | 13,630 | 15,060 |
| Amazon-Computer | 66,562 | 95,530 | 107,740 |
| Amazon-Photo | 11,197 | 37,207 | 45,219 |
| ogbn-arxiv | 291,656 | 392,895 | 457,954 |

### E.6 DATA HETEROGENEITY

Table 8 and 9 shows the data heterogeneity with different numbers of clients in non-overlapping and overlapping settings. To measure the data heterogeneity, we calculate the average cosine distances of label distributions between all pairs of subgraphs. We can observe that a larger number of clients leads to more severe data heterogeneity and the non-overlapping setting is more heterogeneous than the overlapping setting.

### E.7 TIME COST OF PREPROCESSING

Table 10 shows the average time of preprocessing including calculating the PPR matrix and positional encoding for each client on different datasets. The time cost is acceptable considering the PPR matrix and positional encoding only need to be computed once before training.

### E.8 PRIVACY BUDGET OF LDP

Table 8: Data heterogeneity measured by the average cosine distance of the label distributions between all pairs of subgraphs in the non-overlapping setting.

| Dataset | 5 clients | 10 clients | 20 clients |
|---|---|---|---|
| Cora | 0.574 | 0.638 | 0.689 |
| CiteSeer | 0.527 | 0.566 | 0.576 |
| PubMed | 0.345 | 0.362 | 0.400 |
| Amazon-Computer | 0.544 | 0.616 | 0.654 |
| Amazon-Photo | 0.664 | 0.718 | 0.759 |
| ogbn-arxiv | 0.593 | 0.683 | 0.724 |

Table 9: Data heterogeneity measured by the average cosine distance of the label distributions between all pairs of subgraphs in the overlapping setting.

| Dataset | 10 clients | 30 clients | 50 clients |
|---|---|---|---|
| Cora | 0.291 | 0.549 | 0.643 |
| CiteSeer | 0.338 | 0.537 | 0.566 |
| PubMed | 0.238 | 0.370 | 0.382 |
| Amazon-Computer | 0.368 | 0.570 | 0.618 |
| Amazon-Photo | 0.308 | 0.689 | 0.721 |
| ogbn-arxiv | 0.427 | 0.654 | 0.687 |

In local differential privacy, denote the input vector of model parameters or representations as $g$, the LDP mechanism as $\mathcal{M}$, and the clipping threshold as $\delta$, we have: $\mathcal{M}(g) = clip(g, \delta) + n$, where $n \sim Laplace(0, \lambda)$ is the Laplace noise with 0 mean and noise strength $\lambda$. Figure 8 shows the privacy budget $\epsilon$ of the LDP module, which is calculated by $\epsilon = \frac{2\delta}{\lambda}$ following previous works (Wu et al., 2021; Qi et al., 2020). We can observe that larger $\lambda$ and smaller $\delta$ lead to stricter privacy protection (smaller $\epsilon$).

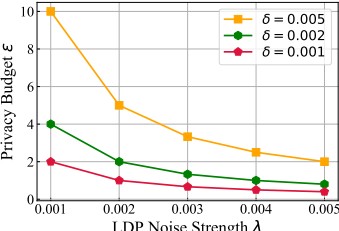

Figure 8: Privacy budget $\epsilon$ of the LDP.

### E.9 ABLATION STUDIES

To analyze the contribution of each module, we conduct ablation studies by removing the global attention module and personalized aggregation module respectively in FedGT. As shown in Figure 9, we observe that the performance of FedGT obviously overperforms its two variants, demonstrating the effectiveness of the designed modules. Moreover, the benefits brought by each module may vary across different datasets. In particular, personalized aggregation is more important on smaller datasets such as Cora and CiteSeer. This may be explained by the fact that clients lack local training data and can only boost performance by identifying relevant clients for joint training. On the contrary, FedGT benefits more from global attention on larger datasets such as ogbn-arxiv. This may be because global attention helps capture the long-range dependencies on these large graphs.

### E.10 T-SNE VISUALIZATION OF GLOBAL NODES.

In Figure 9, we select two clients in the non-overlapping setting and visualize the node embeddings as well as the global nodes. We observe that the node embeddings form clusters and the global nodes can occupy the centers of clusters in each client. Therefore, the global nodes can effectively capture the data distribution of the corresponding local subgraph.

### E.11 UNBALANCED PARTITION SETTINGS

Here, we further explore scenarios in which clients have a varying number of nodes, with some having significantly more and others noticeably fewer. We create such a setting based on the non-overlapping setting with 10 clients. For each client, we randomly subsample 10% to 100% of the original nodes.

Table 10: The average time (seconds) of preprocessing including calculating the PPR matrix and positional encoding for each client in the non-overlapping setting.

| Dataset | 5 clients | 10 clients | 20 clients |
|---------|-----------|------------|------------|
| Cora | 0.24 | 0.07 | 0.03 |
| CiteSeer | 0.17 | 0.08 | 0.02 |
| PubMed | 5.36 | 2.17 | 0.76 |
| Amazon-Computer | 3.94 | 1.43 | 0.16 |
| Amazon-Photo | 6.31 | 1.52 | 0.39 |
| ogbn-arxiv | 96.98 | 53.14 | 29.65 |

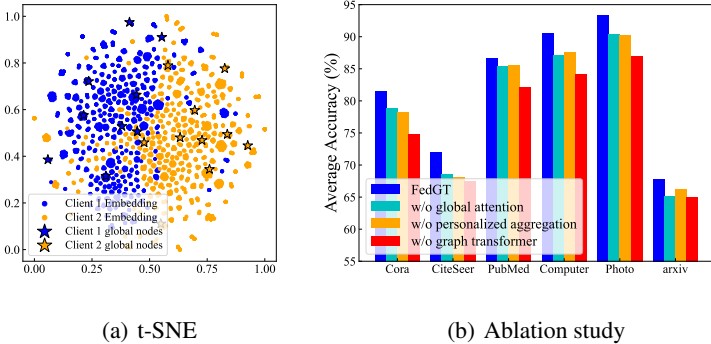

(a) t-SNE                    (b) Ablation study

Figure 9: (a) t-SNE visualization of the node embeddings (dots) and global node (stars) on the Cora dataset and in the non-overlapping setting (10 clients). Different colors indicate different two clients. (b) compares FedGT with its two variants in the non-overlapping setting with 10 clients.

The number of nodes for each client in the default setting (uniform) and the unbalanced setting are shown in Figure. 10. We show the corresponding node classification results in Table. 11. We compare FedGT with the runner-up baseline FED-PUB and observe that our FedGT is robust to the distribution shift and can achieve better results than competitive baselines.

Table 11: Node classification results with the default/unbalanced settings and 10 clients. We report the means and standard deviations over three different runs (%). The best results of each setting are bolded.

| Model | Cora | CiteSeer | PubMed | Computer | Photo | ogbn-arxiv |
|-------|------|----------|--------|----------|-------|------------|
| FED-PUB (default) | 81.45±0.12 | 71.83±0.61 | 86.09±0.17 | 89.73±0.16 | 92.46±0.19 | 66.35±0.16 |
| FedGT (default) | **81.49**±0.41 | **71.98**±0.70 | **86.65**±0.15 | **90.59**±0.09 | **93.17**±0.24 | **67.79**±0.11 |
| FED-PUB (unbalanced) | 73.51±0.40 | 64.32±0.81 | 79.44±0.56 | 85.69±0.48 | 84.87±0.42 | 62.46±0.25 |
| FedGT (unbalanced) | **76.46**±0.35 | **66.72**±0.77 | **83.29**±0.48 | **86.47**±0.55 | **86.74**±0.36 | **65.03**±0.36 |

## F   MORE DISCUSSIONS ON SIMILARITY ESTIMATION

In FedGT, we do not directly use label distribution similarity since uploading label distribution may lead to an additional risk of privacy breach. For example, a group of banks may collaboratively train fraud/anomaly account detection models using subgraph federated learning. In such a case, label distributions can reflect the percentage of fraud/anomaly accounts of the bank and may undermine the bank's reputation if the percentage is large. The privacy issue on sharing label distributions becomes more severe in class imbalance and heterogeneous scenarios. For example, let's assume that most of the nodes in the global graph have label 0 and only several minority nodes have label 1. Then, the server can know which clients have minority nodes by examining the uploaded label distributions, which directly undermines users' privacy.

On the contrary, the global nodes are highly condensed vectors, which can hardly be used to infer private information. Moreover, privacy protection is further guaranteed by applying the local differential privacy scheme.

| (a) Cora | (b) Citeseer | (c) PubMed | (d) Computer | (e) Photo | (f) ogbn-arxiv |

Figure 10: The number of nodes for each data in the default setting (uniform) and the unbalanced setting.

Table 12: Results of FedGT with different similarity estimation methods for personalized aggregation (%). The overlapping setting and the Cora dataset are used here.

| Dataset | 10 clients | 30 clients | 50 clients |
|---|---|---|---|
| FedAvg | 77.82 | 71.40 | 68.79 |
| Local update | 78.36 | 74.71 | 70.58 |
| Label distribution | **81.79** | 77.45 | 75.67 |
| FedGT w/o optimal transport | 79.80 | 76.19 | 75.16 |
| FedGT | 81.73 | **77.84** | **76.20** |

To further explore the influence of similarity estimation, we compare the experimental results with different similarity estimation methods in the overlapping setting on Cora in Table 12. These methods include the similarity of local updates, label distributions, and the global nodes used in FedGT. We also show the results of FedGT using FedAvg as the aggregation scheme for reference.

In Table 12, we observe that FedGT can achieve better results than Label distribution in scenarios with a larger number of clients. It is probably due to the reason that FedGT can dynamically adjust the similarity matrix along with model training, which is more important in cases with less local data. Specifically, at the beginning of FL training, the weighted averaging in FedGT is similar to FedAvg due to the random initialization of global nodes, which helps clients firstly collaboratively learn a general model with distributed data. With the training of FL, the similarity matrix of FedGT gradually forms clusters, which helps train personalized models for each client. On the contrary, the similarity matrix of label distribution is fixed and cannot adjust according to the training of FL. Therefore, the personalized aggregation based on the similarity of aligned global nodes is more suitable for FedGT.

## G    COMPLEXITY ANALYSIS OF FEDGT

Algorithm. 2 and 3 show the pseudo algorithms of FedGT. As for clients, the PPR matrix and positional encoding only need to be computed once before training and will not bring much burden (Appendix E.7). The proposed Graph Transformer has a linear computational cost of $O(n(n_g + n_s))$. In the server, the extra overhead mainly comes from similarity computation $O(n_g^3)$, which is discussed in Section 4.2. Besides transmitting model updates, the transmission of global nodes brings an extra cost of $O(n_g d)$, which is also acceptable since $n_g$ and $d$ are small constants. Overall, the computational and communication overhead of FedGT is acceptable.

## H    LIMITATIONS AND FUTURE WORKS

There are several potential directions to further improve our work. Firstly, FedGT tackles the missing links issue implicitly with the powerful graph transformer architecture. We have empirically verified that FedGT is robust to missing links (e.g., the classification accuracy only drops from 90.78% to 90.22% when the number of clients increases from 5 to 20 and the number of missing links increases by 54,878 on Amazon-Computer). In the future, we may combine explicit missing link recovery methods to further address the missing link challenge. Another limitation of FedGT is that only data heterogeneity between subgraphs/clients is considered. We are aware that there are also data heterogeneity problems within the subgraph. In the future, we will consider the data heterogeneity within subgraphs in our model design. Finally, we plan to explore other positional encodings (PE)

in the future. To summarize, we believe our FedGT has shown promising results in tackling the challenges in subgraph FL. We will further improve our work to benefit the broad community.

