# OpenReview forum: "FedGT: Federated Node Classification with Scalable Graph Transformer"
_ICLR.cc/2024/Conference — Submitted to ICLR 2024_

### Official Review · Reviewer_nEXk · 2023-10-24

**Soundness:** 3 good
**Presentation:** 3 good
**Contribution:** 3 good
**Rating:** 8
**Confidence:** 4

**Summary:**

In this paper, the authors propose a scalable Federated Graph Transformer
(FedGT) to address the data heterogeneity and missing link challenges. In contrast to GNNs that follow message-passing schemes and focus on local neighborhoods, Graph Transformer has a global receptive field to learn long-range dependencies and is, therefore, more robust to missing links. Moreover, a novel personalized aggregation scheme is proposed. Extensive experiments show the advantages of FedGT over baselines in 6 datasets and 2 subgraph settings.

**Strengths:**

1.The paper is well-written and organized. The details of the models are described clearly and are convincing.
2.The limitations of applying GNNs for subgraph federated learning are clearly illustrated in Figure 1 and Figure 4 in appendix. The motivation for leveraging graph transformers is easy to understand.
3.The authors proposed a series of effective modules to tackle the challenges, including scalable graph transformers, personalized aggregation, and global nodes. The contribution is significant enough.
4.FedGT is compared with a series of SOTA baselines, including personalized FL methods, federated graph learning methods, and adapted graph transformers. Extensive experiments on 6 datasets and 2 subgraph settings demonstrate
that FedGT can achieve state-of-the-art performance.

**Weaknesses:**

1.The authors are suggested to clearly discuss the case studies in the main paper.
2.Leveraging local differential privacy mechanisms to protect privacy in FL is not new.
3.Please provide more explanations of the assumptions in Theorem 1.

**Questions:**

1.Can the authors introduce more about the roles of global nodes in FedGT?
2.Is FedGT applicable to other subgraph settings?

---

> ### Author Response · Authors · 2023-11-16
> **Response to Reviewer nEXk**
>
> Thanks for your detailed comments and appreciation!
> **Q1**: The authors are suggested to clearly discuss the case studies in the main paper.
> **R1**: Thanks for the suggestion! Due to the limit of the main paper, we provided some case studies such as the t-SNE visualizations of node embeddings and global nodes in Figure 9. We also provided more experimental results in Appendix. E. We will bring the case studies and more experiment results to the main paper in the final version.
>
> **Q2**: Leveraging local differential privacy mechanisms to protect privacy in FL is not new.
> **R2**: Local differential privacy is not our main contribution in FedGT. Following previous works [1,2], local differential privacy is added to further protect privacy. Compared with previous works on subgraph federated learning (e.g., the baseline methods we compare in our paper), FedGT has comparable or better privacy protection levels.
>
> [1] Chuhan Wu, Fangzhao Wu, Yang Cao, Yongfeng Huang, and Xing Xie. Fedgnn: Federated graph
> neural network for privacy-preserving recommendation. KDD, 2021
> [2] Tao Qi, Fangzhao Wu, Chuhan Wu, Yongfeng Huang, and Xing Xie. Privacy-preserving news
> recommendation model learning. EMNLP, 2020.
>
> **Q3**: Please provide more explanations of the assumptions in Theorem 1.
> **R3**: The assumptions in Theorem 1 are mainly two-fold: (1) Firstly, we assume the attention score function is Lipschitz continuous with constant C. We follow this assumption used in previous works e.g., [3]. Moreover, as indicated by [4], the self-attention is Lipschitz if the inputs are bounded and compact. In the data preprocessing, the input node features are normalized. Therefore, the assumption is satisfied in experiments; (2) Secondly, we assume nodes are equally distributed to the global node. In our experiments, we observe that the clustering in FedGT is not sensitive to graph partitioning and the nodes generally distribute evenly to the global nodes. For example, we probed the distribution of nodes to 10 global nodes in a client on the Cora dataset. There are a total 247 nodes in the client and the number of assigned nodes to global nodes is (27, 28, 22, 24, 25, 22, 21, 24, 25, 29), which is quite close to the uniform distribution. To further satisfy the assumption, we can use balanced clustering that ensures nodes are equally distributed to the global nodes. Therefore, these assumptions are reasonable for FedGT.
>
> [3]Kezhi Kong et al., GOAT: A Global Transformer on Large-scale Graphs, ICML 2023
> [4]Kim, Hyunjik et al., The lipschitz constant of self-attention. ICML, 2021.
>
> **Q4**: Can the authors introduce more about the roles of global nodes in FedGT
> **R4**: The global nodes play critical roles in FedGT. Firstly, in the hybrid attention, each node attends to the sampled neighbors in the local subgraph and a set of global nodes representing the global context. As discussed in Sec.4.1 and the experiments, global nodes can supplement the missing information of missing links. Secondly, the personalized aggregation scheme calculates the similarity scores based on global nodes from different clients because global nodes can reflect the corresponding data distribution of local subgraphs. Therefore, global nodes serve as the nexus between hybrid attention and personalized aggregation.
>
> In FedGT, the global nodes are dynamically updated with an online clustering algorithm (Algorithm 1). The global nodes can be regarded as the cluster centroids of node representations and reflect the overall data distribution of the local subgraph.
>
> **Q5**: Is FedGT applicable to other subgraph settings
> **R5**: Thanks for the questions! In FedGT, we mainly follow the setting of previous works such as [1]. In the future, we may explore other settings such as clients with varying numbers of nodes. Due to the limit time of rebuttal, we will incorporate systematic studies in the future version.

---

> > ### Comment · Reviewer_nEXk · 2023-11-23
> > **Response to authors**
> >
> > Thanks the authors for the detailed rebuttal. Most of my concerns are well addressed. As noted by all the reviewers, the idea of leveraging graph transformer architecture for federated graph learning is both novel and well-motivated. Therefore, I recommend accepting this paper, which will be an important basline for federated graph learning area.

---

### Official Review · Reviewer_uFDb · 2023-10-30

**Soundness:** 2 fair
**Presentation:** 2 fair
**Contribution:** 2 fair
**Rating:** 5
**Confidence:** 4

**Summary:**

This paper proposes a scalable Federated Graph Transformer (FedGT) for subgraph federated learning, which addresses the challenges of missing links between subgraphs and subgraph heterogeneity. It uses a hybrid attention scheme to reduce complexity while ensuring a global receptive field and computes clients’ similarity for personalized aggregation.

**Strengths:**

1.	The paper is easy to read, and generally well-written.
2.	The idea of using the Graph Transformer to address the issue of missing links across clients is well-motivated.

**Weaknesses:**

1.	How to aggregate global nodes is not clearly illustrated. On page 6, the authors state, “the global nodes are first aligned with optimal transport and then averaged similar to Equation 8”. However, it is unclear which optimal transport method is applied and how the similarity between global nodes from different clients is calculated. The authors should clarify whether the normalized similarity α_ij used for model parameters is also employed for global nodes or if a different similarity calculation is used. Besides, in Algorithm 3 lines 11 and 13, the aligning process for the global nodes seems to be performed twice, which needs a clearer explanation.

2.	Since the weighted averaging of local models, i.e., Equation (8), is the same in [1], the authors should provide a discussion or experiment to explain why their similarity calculation is superior to that in [1].

3.	To show the convergence rate, Figure 5 and Figure 6 did not contain FED-PUB, which is the runner-up baseline in most cases.

4.	In the ablation study, the authors only conduct experiments on w/o global attention and w/o personalized aggregation. Results of w/o the complete Graph Transformer (i.e., without local attention) should also be provided.
[1] Baek J, Jeong W, Jin J, et al. Personalized subgraph federated learning[C]//International Conference on Machine Learning. PMLR, 2023: 1396-1415.

**Questions:**

1.	The authors opt for a consistent number of global nodes n_g across all clients. However, how does the methodology account for scenarios in which clients have a varying number of nodes, with some having significantly more and others noticeably fewer? Is there a suggested approach for determining varying n_g values that are customized to each client’s node count?

2.	In the typical federated learning framework, the number of training samples is considered when aggregating the model parameters. However, Equation (8) only uses the normalized similarity for the weighted aggregation. Why can we ignore the number of training samples here? Or do we assume the number of training samples is equivalent across clients?

3.	The Hungarian algorithm only finds a bijective mapping while optimal transport can be generalized to many-to-many cases, could the authors explain the reason for making a one-to-one alignment of the global nodes?

4.	Since the global nodes are dynamically updated during the training, and the representations of the nodes are not stable at the beginning of the training, would this impact the effectiveness of similarity calculation based on the global nodes?

---

> ### Author Response · Authors · 2023-11-16
> **Response to Reviewer uFDb**
>
> Thanks for your valuable questions and suggestions!
> **Q1**: How to aggregate global nodes is not clearly illustrated. On page 6, the authors state, “the global nodes are first aligned with optimal transport and then averaged similar to Equation 8”. However, it is unclear which optimal transport method is applied and how the similarity between global nodes from different clients is calculated. The authors should clarify whether the normalized similarity α_ij used for model parameters is also employed for global nodes or if a different similarity calculation is used.
> Besides, in Algorithm 3 lines 11 and 13, the aligning process for the global nodes seems to be performed twice, which needs a clearer explanation.
>
> **R1**: Thanks for the question. For aggregating the global nodes, the optimal transport algorithm in Equation 7 is used to align the global nodes and calculate similarity. The normalized similarity α_ij used for model parameters is also employed for global nodes.
> We will state this more clearly in our revised paper.
>
> As for Algorithm 3 lines 11 and 13, the aligning process is only done once in our algorithm. We restate “Obtain $\hat{\mu}_i^{(r)}$ with aligning and weighted averaging.” to emphasize the global nodes are first aligned and then averaged. We will add more explanations along with the algorithm.
>
> **Q2**: Since the weighted averaging of local models, i.e., Equation (8), is the same in [1], the authors should provide a discussion or experiment to explain why their similarity calculation is superior to that in [1].
>
>  [1] Baek J, Jeong W, Jin J, et al. Personalized subgraph federated learning[C] International Conference on Machine Learning. PMLR, 2023: 1396-1415.
>
> **R2**: Thanks for the question! The main difference between our weighted averaging and [1] is how to calculate the similarity between clients. [1] proposes to use functional similarity to measure the similarity between clients. However, the quality of the calculated similarity is influenced by the randomly generated input graph (stochastic block model used in [1]). The situation becomes worse when the generated input graph has a different distribution from the original data on each client.
> Moreover, the generation and transmission of the generated input graph induces extra computation and communication costs.
>
> In our paper, we use the global nodes from each client to calculate the similarity between clients. The global nodes are intrinsic to FedGT and can well reflect the data distribution of each client. The calculated similarity is more stable and not influenced by other artifacts. Therefore, our similarity calculation is superior to that in [1].
>
> We will add these explanations to our paper.
>
> **Q3**: To show the convergence rate, Figure 5 and Figure 6 did not contain FED-PUB, which is the runner-up baseline in most cases.
> **R3**: In Figures 5 and 6, we select several representative baselines to compare with FedGT. For example, FedPer is the representative personalized FL baseline; FedSage+ is the representative subgraph FL baseline; GraphGPS is the representative graph transformer baseline. In the revised version, we added FED-PUB to compare with FedGT. We can observe that FedGT has better convergence performance than the other baselines including FED-PUB.
>
> **Q4**: In the ablation study, the authors only conduct experiments on w/o global attention and w/o personalized aggregation. Results of w/o the complete Graph Transformer (i.e., without local attention) should also be provided.
> **R4**: Thanks for the suggestion! In the revised paper, we include the results of w/o graph transformer in our ablation studies (Figure 9). We observed that the performance of FedGT w/o graph transformer drops a lot, which further verifies the effectiveness of graph transformer in subgraph federated learning.
>
> **Q5**: The authors opt for a consistent number of global nodes n_g across all clients. However, how does the methodology account for scenarios in which clients have a varying number of nodes, with some having significantly more and others noticeably fewer? Is there a suggested approach for determining varying n_g values that are customized to each client’s node count?
> **R5**: Thanks for the question! In our paper, we follow the setting of [1] where the number of nodes of each client is roughly the same. In scenarios where clients have varying numbers of nodes, the number of global nodes n_g can be set proportional to the number of nodes. We will systematically explore such settings in our future works.

---

> > ### Comment · Reviewer_uFDb · 2023-11-22
> > **response**
> >
> > Thanks for the authors’ response. The authors have clarified the aggregation of global nodes and added further experiments on convergence comparison and ablation study. However, as the authors admit, the proposed method’s applicability is limited to scenarios where clients possess the same number of nodes. Consequently, I have reconsidered my evaluation and decided to raise my score to 5.

---

> ### Author Response · Authors · 2023-11-16
> **Response to Reviewer uFDb**
>
> **Q6**: In the typical federated learning framework, the number of training samples is considered when aggregating the model parameters. However, Equation (8) only uses the normalized similarity for the weighted aggregation. Why can we ignore the number of training samples here? Or do we assume the number of training samples is equivalent across clients?
>
> **R6**: Thanks for the question! In our work, we follow the setting of previous works such as [1] where the number of nodes of local graphs is roughly the same. Therefore, we only need to consider the normalized similarity for aggregation and ignore the difference in the number of training samples, which is the same as the aggregation scheme in FED-PUB [1]. In future works, we will systematically explore the influence of the number of nodes and consider it in aggregations.
>
> **Q7**: The Hungarian algorithm only finds a bijective mapping while optimal transport can be generalized to many-to-many cases, could the authors explain the reason for making a one-to-one alignment of the global nodes?
>
> **R7**: In FedGT, the Hungarian algorithm is used to calculate the pairwise similarity for personalized aggregation. We use the Hungarian algorithm for its easy implementation and stable performance. To the best of our knowledge, we did not find a suitable many-to-many optimal transport applicable to our setting. Therefore, we adopt the Hungarian algorithm in FedGT. We may explore other optimal transport algorithms in our future works.
>
> **Q8**: Since the global nodes are dynamically updated during the training, and the representations of the nodes are not stable at the beginning of the training, would this impact the effectiveness of similarity calculation based on the global nodes?
>
> **R8**: The global nodes are dynamically updated during the training and the performance has fluctuations at the beginning. However, based on the observation, FedGT converges very quickly after several iterations (e.g., Figures 5 and 6). Therefore, the update of global nodes will not impact the effectiveness of similarity calculation.

---

> ### Author Response · Authors · 2023-11-23
> **Further rebuttal with more results on unbalanced settings**
>
> We thank the reviewer for the response and support! We are happy to know that most of your questions are addressed. As for the default setting that clients possess a similar number of nodes, we follow the same setting with previous works on subgraph federated learning [1, 2]. In our revised version, we further add additional results in the unbalanced setting (Appendix E.11, Figure. 10, Table 11). The number of nodes of clients can vary **10 times**. We compared FedGT with the most competitive baseline FED-PUB. The unbalanced setting is more challenging due to fewer nodes and more missing links. However, we can observe that FedGT is **robust to the distribution shifts and can consistently achieve better results than baselines**.
>
> | Model                  | Cora           | CiteSeer       | PubMed         | Computer       | Photo          | ogbn-arxiv     |
> |------------------------|----------------|----------------|----------------|----------------|----------------|----------------|
> | FED-PUB (default)      | 81.45±0.12     | 71.83±0.61     | 86.09±0.17     | 89.73±0.16     | 92.46±0.19     | 66.35±0.16     |
> | FedGT (default)        | **81.49**±0.41     | **71.98**±0.70     | **86.65**±0.15     | **90.59**±0.09     | **93.17**±0.24     | **67.79**±0.11     |
> | FED-PUB (unbalanced)   | 73.51±0.40     | 64.32±0.81     | 79.44±0.56     | 85.69±0.48     | 84.87±0.42     | 62.46±0.25     |
> | FedGT (unbalanced)     | **76.46**±0.35     | **66.72**±0.77     | **83.29**±0.48     | **86.47**±0.55     | **86.74**±0.36     | **65.03**±0.36     |
>
>
> We hope our further results and explanations can address your concerns. We will be very grateful if you consider increasing to positive scores to support our work!
>
> [1] Ke Zhang, Carl Yang, Xiaoxiao Li, Lichao Sun, and Siu Ming Yiu. Subgraph federated learning with missing neighbor generation. In NeurIPS 21
> [2] Jinheon Baek, Wonyong Jeong, Jiongdao Jin, Jaehong Yoon, and Sung Ju Hwang. Personalized subgraph federated learning. In ICML 23

---

### Official Review · Reviewer_jUaw · 2023-10-31

**Soundness:** 3 good
**Presentation:** 3 good
**Contribution:** 3 good
**Rating:** 5
**Confidence:** 5

**Summary:**

The authors propose to use Graph Transformer and optimal-transport-based personalized aggregation to alleviate the fundamental problems in the subgraph federated learning algorithm such as missing links and subgraph heterogeneity.

**Strengths:**

(1) Leverages graph transformer architecture within subgraph FL for the first time in the federated graph learning literature.

(2) The algorithm is compatible with local DP.

(3) Experimentally shows that Transformers are useful for subgraph federated learning.

(4) Theoretical analysis of global attention being able to capture and approximate information in the whole subgraph is provided.

**Weaknesses:**

(1) How Graph Transformer deals with the missing links is unclear.

(2) The assumption that nodes are equally distributed to the global nodes seems unrealistic due to graph partitioning.

(3) Theorem is not rigorous as it is a known fact that more nodes less error [1]

(4) Local LDP does not guarantee privacy for sensitive node features, edges, or neighborhoods on
distributed graphs [2,3]. Using LDP does not reflect an actual privacy guarantee for this case.

[1] Kim, Hyunjik, George Papamakarios, and Andriy Mnih. "The Lipschitz constant of self-attention." International Conference on Machine Learning. PMLR, 2021.
[2] Imola, Jacob, Takao Murakami, and Kamalika Chaudhuri. "Locally differentially private analysis of graph statistics." 30th USENIX security symposium (USENIX Security 21). 2021.
[3]Kasiviswanathan, Shiva Prasad, et al. "Analyzing graphs with node differential privacy." Theory of Cryptography: 10th Theory of Cryptography Conference, TCC 2013, Tokyo, Japan, March 3-6, 2013. Proceedings. Springer Berlin Heidelberg, 2013.

**Questions:**

(1) Could you please compare FedGT with FedDEP [1]?



[1] Zhang, Ke, et al. "Deep Efficient Private Neighbor Generation for Subgraph Federated Learning."

---

> ### Author Response · Authors · 2023-11-16
> **Response to Reviewer jUaw**
>
> We thank you for the detailed reviews and constructive comments!
> **Q1**：How Graph Transformer deals with the missing links is unclear.
> **R1**：We discussed how Graph Transformer deals with missing links in introduction, Section 4, and Appendix B.
> FedGT tackles missing link issues with the powerful graph transformer architecture and the global nodes. Most GNNs follow a message-passing paradigm that is likely to make false predictions with altered or missing links. In contrast, Graph Transformer is robust to missing links due to its global attention scheme. Moreover, the global nodes in FedGT capture the global context and get further augmented with personalized aggregation, which can supplement the missing information of cross-subgraph links. Finally, extensive experiments on 6 datasets demonstrate FedGT can overperform state-of-the-art baselines based on GNNs. As pointed by other reviewers, dealing with missing links with graph transformer is well motivated.
> We will state more clearly in our revised paper.
>
> **Q2**: The assumption that nodes are equally distributed to the global nodes seems unrealistic due to graph partitioning.
> **R2**: In FedGT, each client curates a set of global nodes to approximate the global context of the corresponding subgraph. As in Algorithm 1, the global nodes are updated with an online clustering algorithm as the cluster centers. In our experiments, we observe that the clustering in FedGT is not sensitive to graph partitioning and the nodes generally distribute evenly to the global nodes.
> For example, we probed the distribution of nodes to 10 global nodes in a client on the Cora dataset. There are a total 247 nodes in the client and the number of assigned nodes to global nodes is (27, 28, 22, 24, 25, 22, 21, 24, 25, 29), which is quite close to the uniform distribution.
> To further satisfy the assumption, we can use **balanced clustering** [1,2] that ensures nodes are equally distributed to the global nodes. Therefore, the assumption is realistic for FedGT.
>
> [1]Weibo Lin et al., Balanced clustering: a uniform model and fast algorithm. IJCAI, 2019.
> [2]Bienkowski, Marcin, et al. Improved Analysis of Online Balanced Clustering." International Workshop on Approximation and Online Algorithms. 2021.
>
> **Q3**: Theorem is not rigorous as it is a known fact that more nodes less error [3]
> **R3**: Thanks for mentioning the related work [3]. Generally, our theorem aligns well with the fact that more nodes less error. In Figure 5(b), we show that the accuracy generally increases with more global nodes. Due to the randomness of training and the redundant noise brought by global nodes, the classification may have fluctuations with a larger number of global nodes.
>
> [3] Kim, Hyunjik, George Papamakarios, and Andriy Mnih. "The Lipschitz constant of self-attention." International Conference on Machine Learning. PMLR, 2021.
>
> **Q4**: Local LDP does not guarantee privacy for sensitive node features, edges, or neighborhoods on distributed graphs [2,3]. Using LDP does not reflect an actual privacy guarantee for this case.
> **R4**:
> We thank the reviewer for mentioning the related works [4,5] on the private analysis of graph data. We will cite and discuss them in our paper. In our work, we do not directly upload graph statistics. Instead, we apply LDP to the uploaded model parameters and learned global node representations following previous works [6,7] to protect privacy.
> Moreover, local differential privacy is not our main contribution to FedGT. Compared with previous works on subgraph federated learning (e.g., the baseline methods we compare in our paper), FedGT has **comparable or better privacy protection levels**.
>
> [4] Imola, Jacob, Takao Murakami, and Kamalika Chaudhuri. "Locally differentially private analysis of graph statistics." 30th USENIX security symposium (USENIX Security 21). 2021.
> [5] Kasiviswanathan, Shiva Prasad, et al. "Analyzing graphs with node differential privacy." Theory of Cryptography: 10th Theory of Cryptography Conference, TCC 2013, Tokyo, Japan, March 3-6, 2013. Proceedings. Springer Berlin Heidelberg, 2013.
> [6] Chuhan Wu, Fangzhao Wu, Yang Cao, Yongfeng Huang, and Xing Xie. Fedgnn: Federated graph
> neural network for privacy-preserving recommendation. KDD, 2021
> [7] Tao Qi, Fangzhao Wu, Chuhan Wu, Yongfeng Huang, and Xing Xie. Privacy-preserving news
> recommendation model learning. EMNLP, 2020.

---

> > ### Author Response · Authors · 2023-11-17
> > **Response to Reviewer jUaw**
> >
> > **Q5**: Could you please compare FedGT with FedDEP
> > **R5**: Thanks for mentioning the related work FedDEP. We will cite and discuss it in our paper. We searched on explorer and the only available version of FedDEP is at http://www.cs.emory.edu/~jyang71/files/feddep-workshop.pdf . Meanwhile, no code is provided in the paper. Therefore, it is challenging to reproduce FedDEP in the limited time of rebuttal. Thanks for your understanding. In our paper, we have already compared with **6** representative state-of-the-art baselines including the recent method Fed-PUB. Extensive results on various settings show the superior performance of FedGT.

---

### Author Response · Authors · 2023-11-21
**Kind Reminder of the Rebuttal DDL**

Dear reviewers,

Thanks for your detailed and valuable comments! As the ddl of the rebuttal is close, we would like to know whether our rebuttal has addressed your concerns. We will keep improving our paper if there are further questions.

Bests,
Authors

---

### Meta-Review · Area_Chair_B3Xg · 2023-12-06

**Metareview:**

This paper introduces a scalable Federated Graph Transformer (FedGT) designed to overcome the challenges of missing links and data heterogeneity in subgraph federated learning. Unlike traditional Graph Neural Networks (GNNs) that focus on local interactions, FedGT employs a global receptive field through a hybrid attention mechanism, enhancing its robustness to missing links. Additionally, the paper introduces a personalized aggregation scheme, with experiments demonstrating FedGT's superiority over existing baselines across six datasets and two subgraph configurations.

While the proposed FedGT model shows promising results, there are several weaknesses that need to be addressed:

1. The FedGT model claims to have linear computational complexity. However, the paper lacks experiments on large-scale graph data, which are necessary to validate the method's efficiency and performance.

2. FedGT is designed under the assumption that client nodes are approximately evenly distributed, a condition seldom met in numerous federated environments. Extensive federated scenarios demonstrate highly imbalanced distributions among client nodes in federated graph learning, which could potentially hinder the widespread application of FedGT in real-world settings. The existing supplementary experiments were conducted on smaller datasets. These experiments neither provided a detailed explanation of the distribution of imbalanced dataset data nor included further baseline comparison. This lack of comprehensive analysis is insufficient to prove that FedGT can adapt to extremely imbalanced graph federated environments.

3. Although well-trained virtual nodes can represent graph properties and reconstruct the graph's original structure, local LDP fails to ensure privacy for sensitive node features, edges, or neighborhoods in distributed graphs. Transferring virtual nodes associated with local full graphs to the server might still pose a risk of clients' privacy leakage.

4. In FedGT, the Hungarian algorithm is used to compute pairwise similarities for personalized aggregation. However, this approach does not represent an actual optimal matching scenario. The authors should consider applying more advanced methods to measure similarities between virtual nodes, thereby enhancing the model's accuracy and applicability in diverse federated learning environments.

Based on these weaknesses, we recommend rejecting this paper. We hope this feedback helps the authors improve their work.

**Justification For Why Not Higher Score:**

One reviewer championed this paper; however, after reviewing it myself, I believe the weaknesses (as listed in the meta-review) outweigh its merits.

**Justification For Why Not Lower Score:**

N/A

---

### Decision · Program_Chairs · 2024-01-16

Reject